# An essential role for maternal control of Nodal signaling

**Pooja Kumari[1,2†], Patrick C Gilligan[1†], Shimin Lim[1,3], Long Duc Tran[4], Sylke Winkler[5], Robin Philp[6‡], Karuna Sampath[1,2,3]***

[1]Temasek Life Sciences Laboratory, National University of Singapore, Singapore, Singapore; [2]Department of Biological Sciences, National University of Singapore, Singapore, Singapore; [3]School of Biological Sciences, Nanyang Technological University, Singapore, Singapore; [4]Mechanobiology Institute, National University of Singapore, Singapore, Singapore; [5]Department of Cell Biology and Genetics, Max Planck Institute for Molecular Cell Biology and Genetics, Dresden, Germany; [6]Bioprocessing Technology Institute, A*STAR, Singapore, Singapore

**Abstract** Growth factor signaling is essential for pattern formation, growth, differentiation, and maintenance of stem cell pluripotency. Nodal-related signaling factors are required for axis formation and germ layer specification from sea urchins to mammals. Maternal transcripts of the zebrafish Nodal factor, Squint (Sqt), are localized to future embryonic dorsal. The mechanisms by which maternal sqt/nodal RNA is localized and regulated have been unclear. Here, we show that maternal control of Nodal signaling via the conserved Y box-binding protein 1 (Ybx1) is essential. We identified Ybx1 via a proteomic screen. Ybx1 recognizes the 3' untranslated region (UTR) of sqt RNA and prevents premature translation and Sqt/Nodal signaling. Maternal-effect mutations in zebrafish *ybx1* lead to deregulated Nodal signaling, gastrulation failure, and embryonic lethality. Implanted Nodal-coated beads phenocopy *ybx1* mutant defects. Thus, Ybx1 prevents ectopic Nodal activity, revealing a new paradigm in the regulation of Nodal signaling, which is likely to be conserved.

*For correspondence: karuna@tll.org.sg

†These authors contributed equally to this work

‡Present address: LC MS Solutions, Agilent Technologies, Singapore, Singapore

Competing interests: The authors declare that no competing interests exist.

## Introduction

Nodal factors are secreted signaling proteins of the transforming growth factor-β family, with essential functions in axis formation and germ layer specification during embryonic development in sea urchins, snails, ascidians, frogs, fish, and mammals (*Jones et al., 1995*; *Collignon et al., 1996*; *Erter et al., 1998*; *Feldman et al., 1998*; *Rebagliati et al., 1998*; *Sampath et al., 1998*; *Hudson and Yasuo, 2005*; *Shen, 2007*; *Constam, 2009*; *Grande and Patel, 2009*; *Duboc et al., 2010*). Nodal signaling has also been shown to be important for maintaining human ES cell pluripotency (*James et al., 2005*; *Vallier et al., 2005*). Misregulated Nodal signaling has been found associated with tumor metastases (*Topczewska et al., 2006*). Therefore, understanding the mechanisms that regulate Nodal signaling is crucial.

Nodal signaling is regulated by transcription factors such as DRAP1, FoxH1 and Oct4 (*Sirotkin et al., 2000*; *Cao et al., 2008*). Signal transduction occurs by binding of Nodal ligands to the receptor complex, and activation of downstream Smad effectors (*Shen, 2007*; *Schier, 2009*). Feedback regulation of Nodal signaling is mediated by the Lefty antagonists (*Cheng et al., 2000*; *Meno et al., 2001*; *Branford and Yost, 2002*; *Feldman et al., 2002*). Work in *Xenopus*, zebrafish, and humans showed that Nodal signaling is regulated by miRNAs, but the precise mechanism is unknown (*Schier, 2009*; *Luo et al., 2012*). Nodal signaling is also influenced by secretion, endocytosis, lysosomal degradation, post-translational modifications, and processing of the ligands (*Zhang et al., 2004*; *Shen, 2007*; *Tian et al., 2008*; *Constam, 2009*). Spatially restricted translation of exogenous xCR1 reporters in

**eLife digest** In many organisms, embryonic development is controlled in part by RNAs that are deposited into the egg as it forms inside the mother. These 'maternal RNAs' may localize to particular regions of the egg or embryo, where they are then exclusively translated into protein and carry out their specific function. This helps to establish asymmetry in the developing organism—that is, to produce tissues that will eventually become the top or bottom, front or back, and left or right of the organism.

One such maternal RNA encodes Nodal, a key signaling molecule that is conserved across vertebrate and some invertebrate organisms. In zebrafish, the equivalent RNA is called squint, and plays an important role in embryonic development. The squint RNA deposited by the mother localizes to the dorsal region—the embryo's back—and signals that region to make dorsal tissues, but how squint is regulated is not well understood. Now, Kumari et al. identify a protein that controls the positioning of squint RNA, and find that it can also prevent this RNA from being translated into protein.

The squint RNA contains a 'dorsal localization element' that recruits it to the dorsal cells of the embryo by the 4-cell stage (i.e., within two cell divisions after the egg is fertilized). Kumari et al. identified a protein called Ybx1 that could bind to this element: this protein may help to correctly position RNAs in many other organisms, including fruit flies and mammals. Strikingly, embryos formed abnormally when their maternally derived Ybx1 protein was mutant, and these mutations also prevented the squint RNA from localizing properly. This suggests that maternally derived Ybx1 protein directly regulates the squint RNA.

As well as positioning the squint RNA correctly, the embryo must translate this RNA into protein at the right time. In embryos with mutant maternal Ybx1 protein, the Squint protein could be detected at the 16-cell stage, whereas in wild-type embryos this protein is not translated until the 256-cell stage; this indicates that Ybx1 protein might normally repress the translation of the squint RNA. Indeed, Kumari et al. found that Ybx1 binds to another protein—eIF4E—that recruits mRNAs to the ribosome (the cell's translational machinery). Ybx1 might therefore prevent eIF4E from associating with other components of the ribosomal complex, and initiating the translation of the squint RNA, until additional signals have been received. It will be interesting to determine how widespread this regulatory mechanism is in other organisms.

frogs has been suggested (*Zhang et al., 2009*). But so far, a direct role for translational control in regulation of Nodal signaling has not been uncovered.

We showed previously that maternal RNA encoding the zebrafish Nodal factor, Squint (Sqt), is localized to two cells by the 4-cell stage, and predicts embryonic dorsal (*Gore et al., 2005*). RNA localization is an important mechanism that generates asymmetry in cells and organisms. For example, bicoid RNA localization in *Drosophila* oocytes and embryos is required for specification of anterior cell fates, and localization of maternal pem-1 and macho-1 RNAs determines the posterior end of ascidian embryos (*Nishida and Sawada, 2001*; *Sardet et al., 2003*; *St Johnston and Nüsslein-Volhard, 1992*). Mechanisms to ensure correct transport of the RNA and inhibition of translation until the RNA reaches its destination are essential for this process (*Johnstone and Lasko, 2001*; *Martin and Ephrussi, 2009*). In addition, translational control is an important step for regulation of some RNAs. For instance, a proportion of maternal nanos RNA is uniformly distributed in the cytoplasm of *Drosophila* embryos but is not translated, and Nanos protein is only synthesized from localized nanos RNA at the posterior pole (*Gavis and Lehmann, 1994*; *Smibert et al., 1996*; *Bergsten and Gavis, 1999*; *Crucs et al., 2000*). In zebrafish embryos, transport of maternal sqt/nodal RNA to future dorsal is dependent on the microtubule cytoskeleton (*Gore et al., 2005*). However, how maternal sqt RNA is regulated until it reaches future dorsal was not known.

To understand global regulation of sqt/nodal we carried out a screen for sqt 3'UTR-binding proteins, and show here, that the conserved Y box-binding protein 1 (Ybx1) binds the 3' untranslated region (UTR) in sqt RNA. Genetic analysis of *ybx1* mutants shows that maternal Ybx1 function is essential for embryonic development. Loss of Ybx1 function causes mis-localization of sqt RNA and precocious Sqt protein translation, leading to premature and uncontrolled Nodal signaling, and embryonic lethality.

Thus, maternal Ybx1 is required for translational control of Nodal signaling. Since the 3′UTR of mammalian nodal RNAs can localize in fish embryos, it is likely that this control mechanism of translational repression is conserved. Our results identify a new mode of regulation of Nodal signaling, and highlight the role of maternal factors in regulation of growth factor signaling and cell-type specification in vertebrates.

## Results

### Identification of a dorsal localization element (DLE)-binding factor in zebrafish embryos

The dorsal localization element (DLE) of sqt RNA resides in the first 50 nucleotides of the sqt 3′UTR, and encompasses sequence and structural elements (*Gilligan et al., 2011*). To identify the proteins that specifically recognize the DLE, 100-nucleotide long radioactive probes spanning the sqt 3′UTR were used for RNA gel-shift assays with zebrafish whole embryo extracts (*Figure 1A,B*). We observed a number of binding activities in gel-shift assays with sqt probes (*Figure 1B*). The DLE-containing sqt1 probe was bound by an activity, which we named sqt-RNA Binding Factor 1 (SRBF1; arrow in *Figure 1B*). Competition gel-shift assays with control gfp, vg1 and cyclops RNA show that SRBF1 preferentially binds the sqt DLE (*Figure 1C,D*). RNA-cross-linking assays show that SRBF1 is approximately 48–50 kDa (*Figure 1—figure supplement 1*). To precisely map the SRBF1 binding site, a 10-nucleotide sqt1 deletion series was tested for binding. Whereas deletions in the coding sequence did not affect SRBF1 binding, deletions 1–4 (Δ1–Δ4, *Figure 1C,E*) abolish, or significantly reduce binding to the sqt1 probe. The SRBF1 binding site overlaps with sequences required for dorsal localization of sqt RNA (i.e., Δ1 and Δ2; [*Gilligan et al., 2011*], and *Figure 1C,E*). Thus, SRBF1 is the activity that binds to the sqt DLE.

### SRBF1 is the conserved nucleic acid binding protein, Y box-binding protein 1 (Ybx1)

To identify SRBF1, we purified it by column chromatography, and screened individual fractions by gel mobility-shift and RNA cross-linking assays (*Figure 2A*). A 48 kDa factor that co-fractionated with SRBF1 activity (*Figure 2B,C*) was identified by mass spectrometry to be the conserved nucleic acid binding protein, Y box-binding protein 1 (Ybx1). Ybx1 contains a 'cold shock' domain (CSD; *Figure 2— figure supplement 1A,B*), similar to bacterial cold shock protein CspA (*Eliseeva et al., 2012*). Ybx1 homologs are associated with localized RNAs in *Drosophila, Ciona,* and *Xenopus,* and mutations in mouse Ybx1 cause lethality (*Bouvet et al., 1995*; *Wilhelm et al., 2000*; *Tanaka et al., 2004*; *Eliseeva et al., 2012*). Ybx1 contains a conserved actin binding domain (ABD), also found in *Drosophila* Ypsilon schachtel (Yps) (*Figure 2—figure supplement 1A*). The dimerization domain (DD) and non-canonical nuclear localization signal (NLS) are conserved amongst the vertebrate Ybx1 proteins (*Figure 2— figure supplement 1A*) (*Eliseeva et al., 2012*).

Ybx1 is an abundant RNA-binding protein, with many functions. Therefore, to confirm if Ybx1 is SRBF1, *ybx1* cDNA sequences were cloned, and recombinant Ybx1 (rYbx1) was tested for sqt DLE-binding in gel-shift assays. Embryonic SRBF1 and rYbx1 bind to sense sqt1 RNA, but not to control gapdh, or to antisense sqt1 probes (*Figure 2D*). Control *Escherichia coli* lysate did not bind to sqt1 or gapdh probes, and recombinant zebrafish Lin28A, that also contains a cold shock domain (*Moss et al., 1997*), did not bind the sqt DLE (*Figure 2D* and *Figure 2—figure supplement 2A*). Semi-quantitative competition gel-shift assays show that embryonic SRBF1 and rYbx1 bind to sqt DLE sequences with the same specificity (*Figure 2—figure supplement 2B*). However, rYbx1 does not bind to the UTRs of other localized RNAs, such as vg1 and wnt8a (*Figure 2—figure supplement 2C*). To determine if Ybx1 forms protein–RNA complexes in vivo with sqt RNA, we performed RNA-immunoprecipitation (RNA-IP) with embryo lysates, and RT-PCR to detect sqt. RNA-IP with anti-Ybx1 antibodies shows a sqt product, whereas control (RT-) and RNA-IP using IgG antibodies do not show any product (*Figure 2E*). Under the same conditions, gapdh and wnt8a RNA are not detected. Therefore, Ybx1 specifically binds to sqt RNA in early embryos, and has all the characteristics of SRBF1.

### The N-terminus of Ybx1 is required for binding sqt RNA

To identify the regions of Ybx1 that bind the sqt DLE sequence and confer specificity to the interactions, we made a series of deletions that removed each of the various domains (single-stranded DNA-binding domain, ssDBD; RNA-binding domain 1,2, RNP1,2; cold shock domain, CSD; dimerization domain, DD; actin binding domain, ABD; nuclear localization signal, NLS) individually, and one

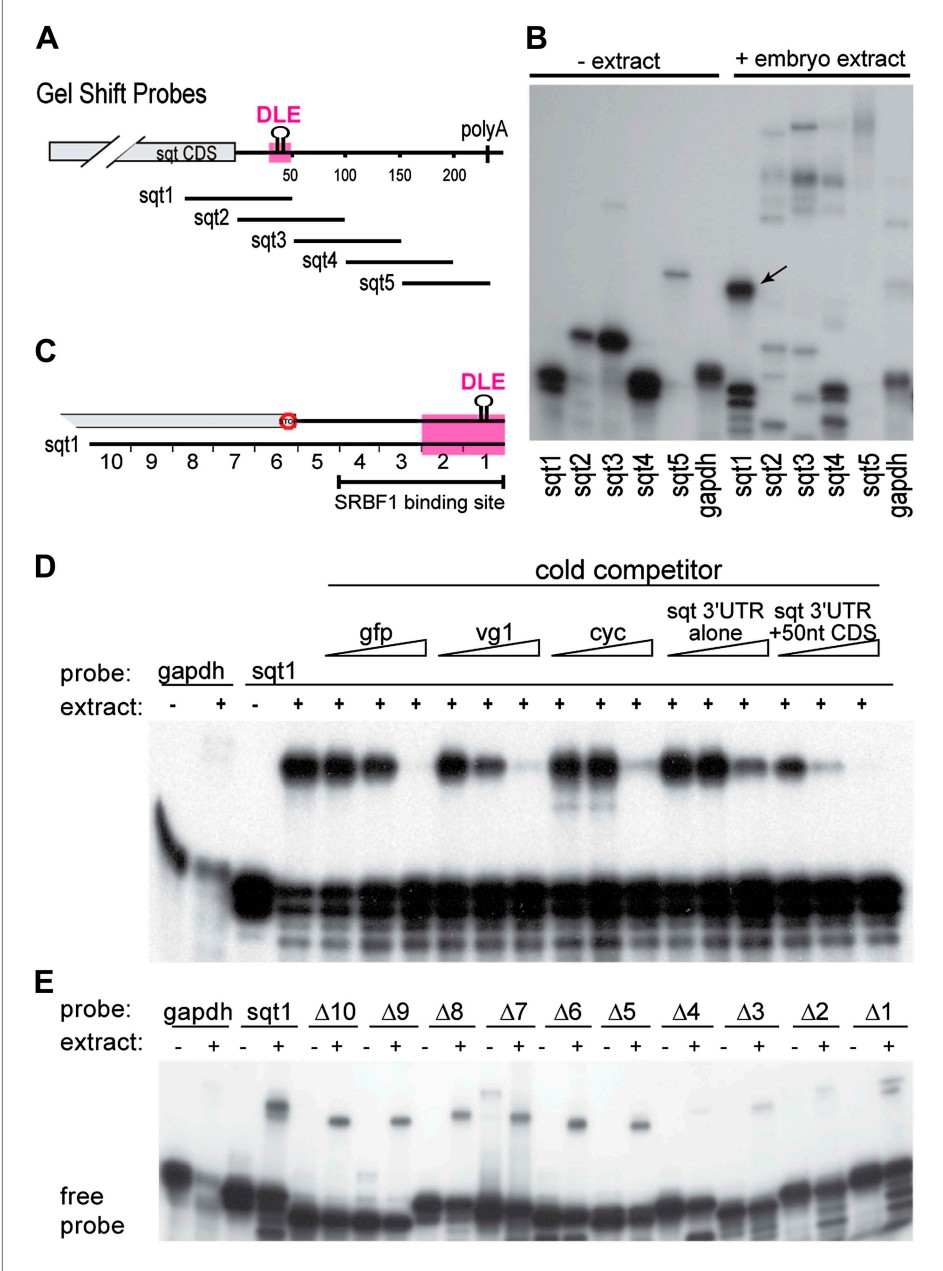

**Figure 1**. SRBF1 binds the sqt Dorsal Localization Element (DLE). (**A**) Schematic of overlapping 100 nucleotide radioactive RNA gel-shift probes spanning the sqt 3'UTR. Position of DLE is highlighted in magenta. (**B**) Autoradiogram showing sqt 3'UTR probes incubated with embryo extract. Several binding activities were observed on the various probes. The 'sqt RNA Binding Factor 1' (SRBF1; black arrow) shift, is detected on the DLE-containing sqt1 probe, and not on other probes. (**C**) Schematic showing the SRBF1 binding site. sqt DLE is highlighted in magenta and the red octagon indicates the stop codon. (**D**) Competition gel-shift assay shows that SRBF1 binds specifically to sqt RNA. The sqt 3'UTR with 50 nucleotides of coding sequence competes more strongly than control gfp, vg1 or cyclops (cyc) RNA for binding to sqt1 probe. Triangles represent 4-fold increases (from 5 ng to 80 ng) of cold competitor RNA. Thus, SRBF1 preferentially binds DLE-sequences. (**E**) The SRBF1 binding site overlaps the DLE. RNA gel-shifts were performed with the sqt1 10 nt deletion series.

The following figure supplements are available for figure 1:

**Figure supplement 1**. SRBF1 is an ~50 kDa protein.

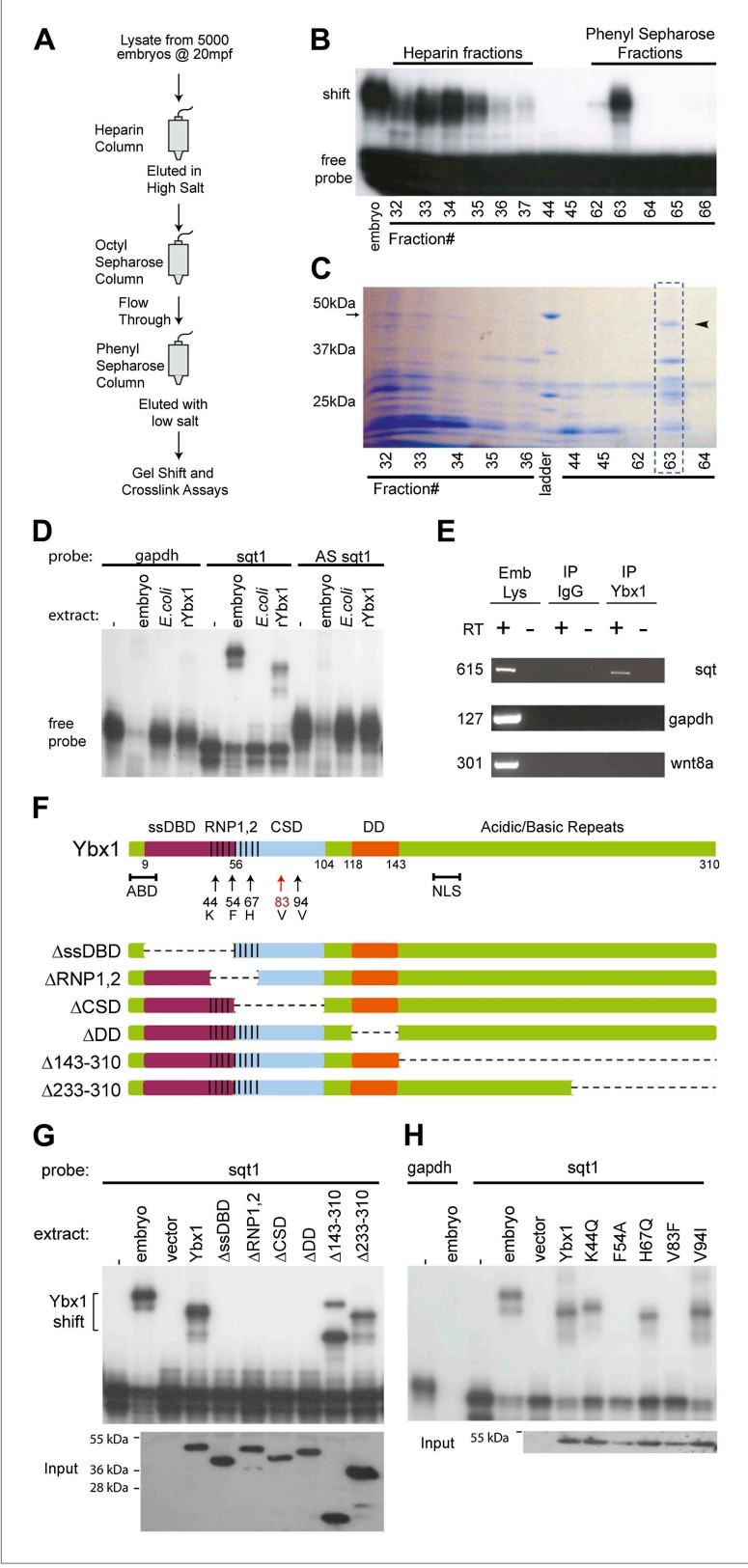

**Figure 2**. SRBF1 contains the nucleic acid binding protein Ybx1. (**A**) Extracts from 5000 embryos collected at 20 mpf were sequentially fractionated on multiple chromatography columns, until SRBF1 was partially pure. At each stage, fractions containing SRBF1 activity were pooled, and loaded onto the next column for further purification. *Figure 2. Continued on next page*

*Figure 2. Continued*

(**B**) A representative native PAGE gel showing SRBF1 purification. Gel-shift analysis of fractions from the heparin and phenyl sepharose columns show SRBF1 activity in fractions 32–37 from heparin and fractions 62–63 from the phenyl sepharose columns. Fractions 33–35 were pooled and added to the hydrophobic columns. Fraction 63 from the phenyl sepharose column contains partially purified SRBF1. (**C**) A Coomassie-blue stained SDS-PAGE gel of the fractions in **B** show a ~48 kDa band that co-fractionates with SRBF1 (black arrowhead in fraction#63). The 48 kDa band from fraction#63 was excised, sequenced by mass spectrometry, and found to contain Ybx1 peptides. (**D**) Gel-shift analysis shows recombinant Ybx1 (rYbx1), similar to embryonic SRBF1, binds to sqt1, but not to control gapdh or antisense sqt1 probes. (**E**) Ybx1 binds sqt RNA in vivo. RT-PCR shows sqt RNA but not control gapdh or wnt8a RNA in RNA-IP with αYbx1 antibodies. Control IgG antibodies do not show any RT-PCR product. RT-PCR from whole embryo lysates is the positive control. PCR product sizes are indicated on the left. (**F**) Schematic diagram showing domain structure of wild-type and mutant Ybx1 proteins. The position of amino acid substitutions is indicated by arrows (V83 in red and all other residues in black). Deletions are indicated by dashed lines. The actin binding domain (ABD), single stranded DNA-binding domain (ssDBD; magenta), RNA-binding domains 1 and 2 (RNP1,2; hashed black lines), Cold shock domain (CSD, blue), dimerization domain (DD; orange), and nuclear localization sequence (NLS) are shown; numbers indicate the amino acid residue. (**G**) Domain analysis of Ybx1. The nucleic acid binding domain (ssDBD, magenta bar in **F**; CSD, blue bar in **F**; RNP1,2, hashed lines in **F**) is required for binding to sqt1, as is the dimerization domain (DD, orange bar in **F**). In contrast, the C-terminus of Ybx1 (Δ143–310 and Δ233–310) is dispensable for binding to sqt1. A western blot with α-His tag antibodies shows expression of the mutant Ybx1 proteins. (**H**) Point mutations in Ybx1 identify key amino acid residues that confer sqt RNA binding. K44, F54, and H67 are expected to contact RNA based upon NMR structure prediction. F54A abolishes binding, whereas H67Q does not affect binding at the protein concentrations shown. V83F abolishes sqt1 binding, whereas V94I binds sqt1. Western blot with α-His tag antibodies shows expression of mutant Ybx1 proteins.

The following figure supplements are available for figure 2:

**Figure supplement 1**. Alignment of Ybx1 sequences shows conservation across species.

**Figure supplement 2**. Ybx1 specifically binds to the sqt 3'UTR.

that removes the entire C-terminal half of the protein (***Figure 2F***). We also made point mutations affecting three amino acid positions (K44Q, F54A and H67Q) that are (1) conserved between bacterial cold shock proteins and Ybx1, (2) shown to be required for RNA binding in bacterial cold shock proteins (***Schröder et al., 1995***; ***Manival et al., 2001***), and (3) suggested by NMR of human Ybx1 to be in contact with RNA (***Kloks et al., 2002***) (***Figure 2—figure supplement 1B***). We find that the C-terminal half of the protein, the ABD, and the NLS are dispensable for sqt RNA binding (***Figure 2F,G***, ***Figure 2— figure supplement 1A*** and data not shown). By contrast, mutations within or overlapping the CSD and RNP domains abolish RNA binding (***Figure 2F–H***). Mutations in the DD also affect binding to RNA. The K44Q, H67Q and V94I mutant proteins were still able to bind the DLE-containing probe at the concentrations shown, whereas the F54A and V83F mutations completely abolished sqt RNA binding (***Figure 2F,H***). These results show that while Ybx1 binds the sqt DLE mainly via its cold-shock domain, other regions such as the DD, RNP and ssDBD are also required for binding to sqt RNA.

## Maternal Ybx1 is essential for early development

Expression of ybx1 RNA is ubiquitous, at all embryonic stages, and western blots show maternal and zygotic Ybx1 protein expression (***Figure 3—figure supplement 1***). To obtain mutants affecting *ybx1*, we generated zinc finger nuclease (ZFN) deletions and screened ENU-induced mutations by TILLING. The *ybx1^sg8^* ZFN allele results in truncated Ybx1 protein lacking the C-terminus (Ybx1^Δ197–310^; ***Figure 3A***). By TILLING, *ybx1^V83F^* (referred henceforth as *ybx1^sa42^*) and *ybx1^V94I^* were identified (***Figures 2F and 3A***). RNA gel-shift assays show that recombinant Ybx1^sa42^ (rYbx1^V83F^) has no detectable DLE-binding, whereas recombinant Ybx1^sg8^ protein (rYbx1^Δ197–310^), which has the RNA-binding CSD, binds the sqt DLE (***Figures 2H and 3B***). Therefore, *ybx1^sa42^* affects the RNA-binding CSD of Ybx1, whereas *ybx1^sg8^* likely encodes a truncated Ybx1 peptide.

Homozygous *ybx1^sa42^* and *ybx1^sg8^* mutant embryos are viable and fertile at 28.5°C, the normal ambient temperature for zebrafish. Extracts of embryos from homozygous *ybx1^sa42^* females (M*ybx1^sa42^*) lack detectable gel-shift activity with sqt1 probes (***Figure 3C***). M*ybx1^sa42^* mutant embryos develop

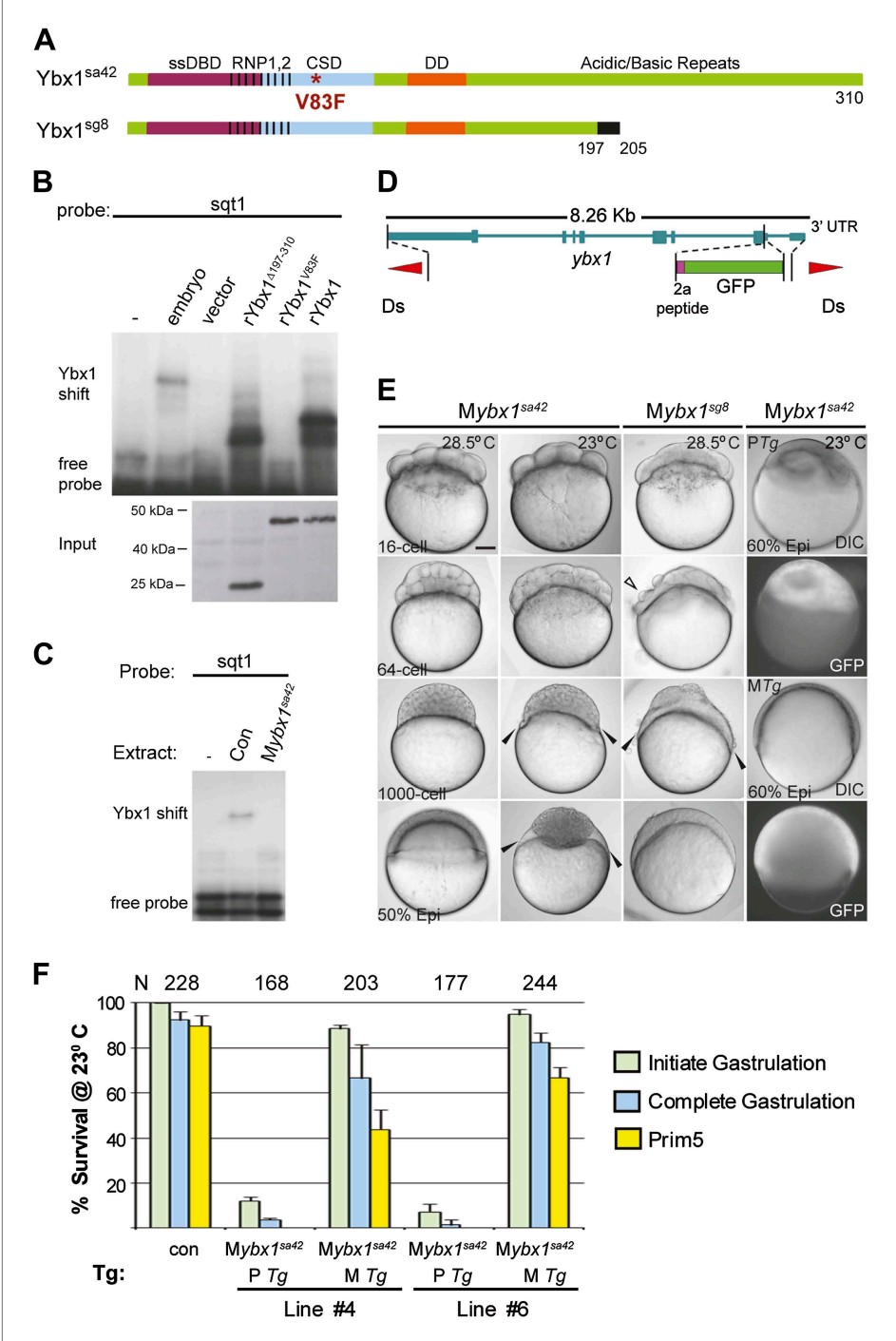

Figure 3. Maternal Ybx1 is essential for early development. (**A**) Schematic of Ybx1 showing the various domains, the V83F mutation in the CSD in *ybx1^sa42^*, and deletion of residues 197–310 in *ybx1^sg8^* mutants. Black box in Ybx1^sg8^ indicates frameshift after residue 197 and premature stop after residue 205. (**B**) rYbx1^V83F^ lacks detectable DLE-binding activity similar to vector control, whereas rYbx1 and rYbx1^Δ197–310^ peptides, and embryo lysates show binding to sqt1 probes. Western blots to detect 6xHis epitope tags show expression of recombinant Ybx1 proteins. (**C**) M*ybx1^sa42^* embryo extracts show no detectable binding to sqt1 probe compared to control extracts. (**D**) Schematic representation of the *ybx1* genomic locus (blue) with positions of viral 2a peptide (magenta bar) and *gfp* (green box) indicated. Red triangles indicate Ds transposon terminal repeats. (**E**) DIC photomicrographs showing 16-cell, 64-cell, 1000-cell and 50% epiboly stage embryos. M*ybx1^sa42^* embryos are viable at 28.5°C. M*ybx1^sg8^* embryos cleave aberrantly after 16-cells (open arrowhead). M*ybx1^sa42^* embryos at 23°C, and M*ybx1^sg8^* embryos fail to initiate

*Figure 3. Continued on next page*

*Figure 3. Continued*

gastrulation, form syncytia (black arrowheads), and arrest. Zygotic Ybx1-GFP expression from P*Tg* does not rescue gastrula arrest in M*ybx1*, whereas maternal Ybx1-GFP expression from M*Tg* leads to normal gastrulation. Scale bar, 100 μm. (**F**) Histogram showing rescue of gastrulation and survival till prim5 stage of M*ybx1* mutants at 23°C by two independent M*Tg* lines (M*Tg* #4 and M*Tg* #6). Some embryos with zygotic expression of ybx1 (P*Tg*) from both lines can initiate gastrulation, but none survive to prim5. Error bars show standard deviation from three experiments. Number of embryos is shown on top of the histogram.

The following figure supplements are available for figure 3:

**Figure supplement 1**. Expression of ybx1 RNA and Ybx1 protein in wild-type embryos.

normally at 28.5°C. However, at 23°C, by early blastula stages, marginal cells in M*ybx1*[sa42] mutants lose cell membranes and a large syncytial layer forms over the yolk (black arrowheads in *Figure 3E*). M*ybx1*[sa42] embryos fail to initiate gastrulation, arrest and do not survive, whereas control embryos from homozygous males (P*ybx1*) survive and develop normally. M*ybx1*[sg8] mutant embryos divide normally till 16-cells, but subsequent cleavages are aberrant, the embryos fail to develop normally, and arrest (*Figure 3E*). Thus, maternal Ybx1 is required for embryonic development.

Injection of ybx1 mRNA into embryos did not rescue M*ybx1* mutants (data not shown). Therefore, we generated stable *ybx1-2a-gfp* transgenic lines harboring genomic y*bx1* sequences fused with viral 2a peptide and GFP sequences (*Figure 3D,E*). Zygotic expression of Ybx1-2a-GFP from paternally inherited *Tg(ybx1-2a-gfp)* transgenes (P*Tg*) did not rescue M*ybx1* mutant embryos (n = 345), although a few embryos initiate gastrulation (*Figure 3E,F*). By contrast, maternal expression of Ybx1-2a-GFP (M*Tg*) from two independent transgenic insertions rescued M*ybx1* mutants (*Figure 3E,F*). M*Tg* expression allowed mutant embryos to undergo gastrulation and survive (*Figure 3F*; n > 200 embryos for each line). Thus, maternal expression of Ybx1 is essential for gastrulation and normal development.

## Ybx1 is required for dorsal localization of sqt RNA

Since Ybx1 was identified as a sqt DLE-binding protein, we examined sqt RNA localization in mutant embryos. In M*ybx1*[sa42] embryos at 28.5°C, there is a lag in sqt RNA transport at the 1-cell stage, but sqt localization is comparable to wild-type and control embryos by 4-cells (*Figure 4A*) (*Gore and Sampath, 2002*; *Gore et al., 2005*). At 23°C, sqt RNA localization in M*ybx1*[sa42] embryos is aberrant at 1-cell and 4-cell stages, forming aggregates in the yolk, that fail to localize to future dorsal cells (*Figure 4A*). Transport of sqt RNA is also disrupted in M*ybx1*[sg8] embryos (*Figure 4—figure supplement 1A*). To determine if Ybx1 functions generally in localization of maternal RNAs, we examined if other maternal transcripts were localized correctly in M*ybx1* mutants. Localization of cortical (vasa, eomesa), axial streamer (snail1a, cyclin B1), and vegetal (wnt8a, grip2) RNAs is unaffected in M*ybx1*[sa42] mutants (*Figure 4A* and *Figure 4—figure supplement 1A,B*). Therefore, Ybx1 does not generally affect all RNA distribution, and amongst the maternal RNAs examined, only sqt localization is disrupted in early M*ybx1* embryos. Maternal Ybx1-2a-GFP expression rescues sqt localization in mutant embryos in contrast to P*Tg* expression (*Figure 4A*). Thus, consistent with Ybx1 binding to the sqt DLE, maternal Ybx1 function is required for sqt RNA localization.

## Ybx1 prevents precocious sqt RNA processing

Ybx1 can function as a transcriptional, post-transcriptional, or translational regulator (*Eliseeva et al., 2012*). To determine whether these processes are affected in mutant embryos, we examined sqt RNA expression (*Figure 4B,C*). Quantitative Real-Time PCR shows that sqt RNA levels are marginally reduced in M*ybx1* mutants compared to controls (*Figure 4B*). In wild-type embryos maternal sqt RNA is non-polyadenylated until the 16-cell stage (*Lim et al., 2012*), but in M*ybx1* embryos, polyA-sqt is detected even at the 1-cell stage, indicating precocious poly-adenylation (*Figure 4C*). We observed un-spliced sqt RNA in control eggs and embryos (*Gore et al., 2007*; *Lim et al., 2012* and *Figure 4C*), in contrast to a previous report (*Bennett et al., 2007*). RT-PCRs to detect sqt intron 1 and intron 2 show that splicing of both introns is accelerated in M*ybx1* embryos compared to controls (*Figure 4C*). These results show that regulated processing of sqt pre-mRNA requires maternal Ybx1 function.

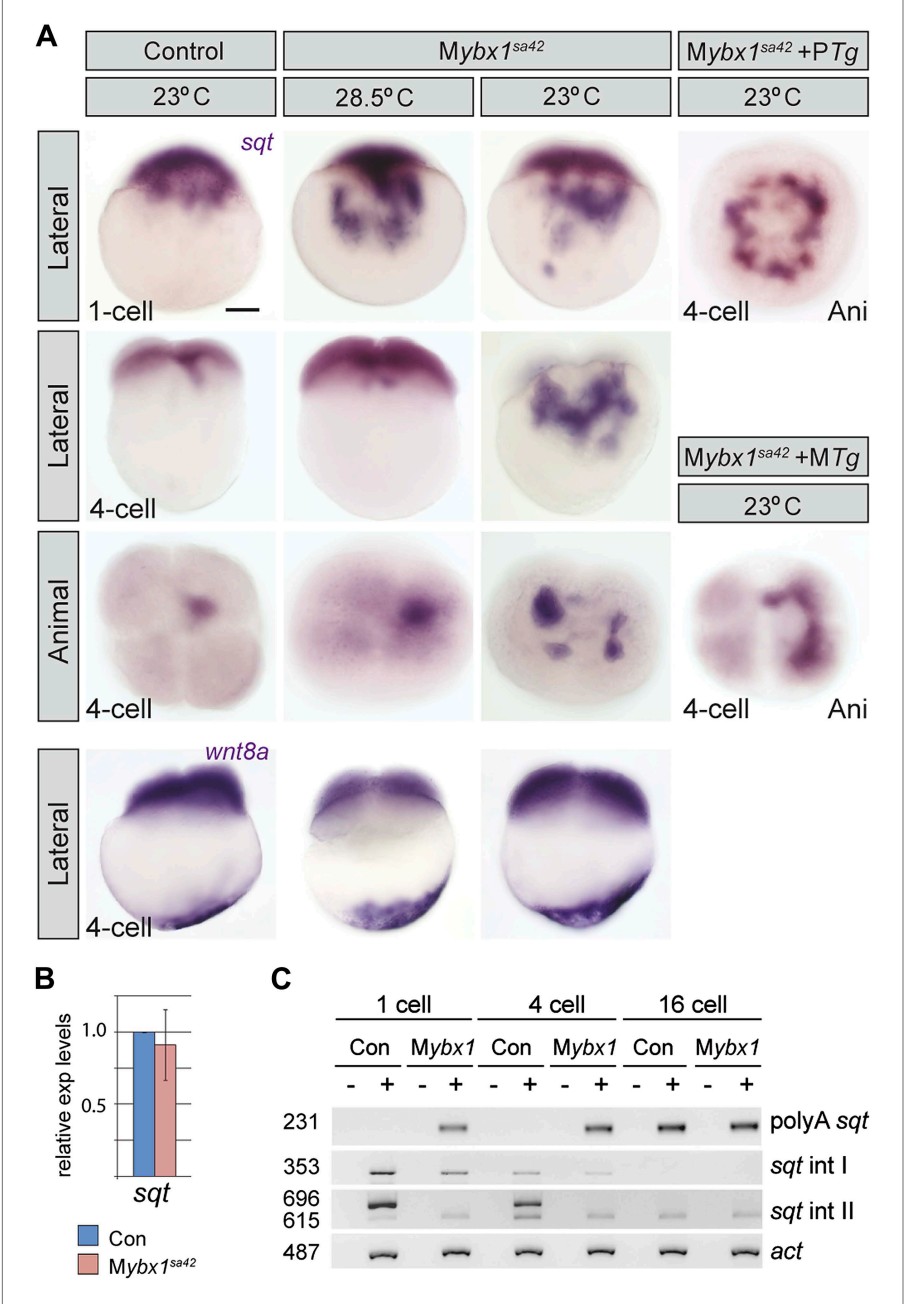

**Figure 4**. Ybx1 is required for localization and regulated processing of sqt RNA. (**A**) Control embryos at 23°C show sqt RNA localization at the 1-cell and 4-cell stage. sqt RNA transport is delayed in 1-cell M*ybx1sa42* embryos at 28.5°C, but localizes correctly by the 4-cell stage. At 23°C, sqt RNA largely remains in the yolk even at 4-cell stage and sqt RNA that reaches the blastoderm is mis-localized. Localization of sqt RNA is restored in M*ybx1* by *ybx1-2a-gfp* M*Tg*, but not by P*Tg*. Localization of wnt8a RNA is normal in M*ybx1sa42* mutants at 28.5°C and 23°C. Scale bar, 100 µm. (**B**) Q-RT-PCRs to detect total sqt RNA levels show a mild reduction in M*ybx1* compared to controls. Error bars show standard deviation from three experiments. (**C**) RT-PCR to detect sqt in control and M*ybx1* mutants at 1-cell, 4-cell and 16-cell stages. Products are indicated on the right, and sizes on the left. Polyadenylated sqt RNA is detected from 16-cells in controls, and at 1-cell in M*ybx1* mutants. Splicing of sqt intron 1 and sqt intron 2 occurs earlier in M*ybx1* embryos compared to controls. PCR to detect *actin* is shown as control.

The following figure supplements are available for figure 4:

**Figure supplement 1**. Localization of sqt RNA is affected in M*ybx1* embryos, whereas vasa, eomesa, snail1a, grip2 and cyclin B1 are unaffected.

## Ybx1 interacts with the eIF4 complex and sqt RNA to repress Sqt translation

To determine if Sqt protein translation is affected in mutant embryos, RNA encoding Sqt-GFP fusion protein was injected, and GFP expression was examined at various stages. Remarkably, Sqt-GFP is observed in 16-cell-stage M*ybx1* embryos, whereas in controls, Sqt-GFP is only detected in blastulae (*Figure 5A*), consistent with the requirement for the Nodal receptors and co-receptor, Oep, from late blastula stages for signaling (*Gritsman et al., 2000*; *Hagos and Dougan, 2007*). Furthermore, Sqt-GFP levels are elevated in M*ybx1* embryos compared to controls (*Figure 5B*). Control GFP and Wnt8a-GFP expression is similar in mutant and control embryos, indicating that translation of other proteins is not affected (*Figure 5—figure supplement 1A,B*). Sqt protein expression is premature in M*ybx1* embryos. Therefore, maternal Ybx1 is required to repress Sqt translation in early embryos.

To determine how Ybx1 regulates translation of sqt RNA, we examined if Ybx1 forms complexes in vivo with translation initiation factors and sqt RNA. Extracts from wild-type embryos were immuno-precipitated with antibodies to Ybx1, eIF4G or eIF4E, followed by western blot, and RT-PCR to detect sqt RNA. Co-immunoprecipitation assays show that Ybx1 interacts with eIF4E but not with eIF4G, and RNA-IP experiments show that sqt RNA is detected in pull-downs with antibodies towards Ybx1, eIF4G and eIF4E. In contrast, gapdh and wnt8a RNA can be detected in RNA-IP with the eIF4G and eIF4E proteins, but not with Ybx1. Hence, Ybx1 binds sqt RNA and the 5' 7-methyl-guanosine cap binding protein eIF4E, but is not found in translation initiation complexes with gapdh and wnt8a RNA (*Figure 5C,D*). Ybx1 has been shown to interact with the 5' cap complex and inhibit translation by displacing eIF4G (*Nekrasov, 2003*). Taken together, these results provide evidence for Ybx1 in regulation of sqt translation by binding to the translation initiation machinery and the 3'UTR of sqt RNA.

## Nodal signaling is premature and elevated in the absence of Ybx1 function

Since Sqt protein is translated prematurely in M*ybx1* mutants, we then determined when Nodal signaling is activated in mutant embryos by examining phosphorylation of the downstream effector, Smad2 (*ten Dijke and Hill, 2004*). Consistent with premature Sqt-GFP translation, endogenous Smad2 is phosphorylated in 64-cell stage M*ybx1* embryos, whereas in control embryos, phospho-Smad2 expression is detected only at late blastula/early gastrula stages (*Figure 6A*). Quantification of phospho-Smad2 levels shows premature and elevated levels of Nodal signaling in M*ybx1* (*Figure 6B*). Thus, Nodal signaling is precociously activated at cleavage stages in mutant embryos.

Analysis of target genes of various signaling pathways shows that expression of Nodal targets (*gsc, ntl, bon* and *sqt*) is increased in M*ybx1* embryos by the 512-cell stage (*Figure 6C*). Expression of the extra-embryonic Yolk Syncytial Layer (YSL) genes, *mxtx2* and *hhex*, is also significantly elevated. By contrast, expression of the Wnt targets *boz* and *vox*, the FGF target *spry4*, and the enveloping layer (EVL) gene *cldE*, is either unchanged or marginally reduced in M*ybx1* mutants compared to controls (*Figure 6C,D*). The YSL expression domain of the Nodal target genes *sqt* and *gsc* is expanded in M*ybx1*, as is YSL expression of *mxtx2*, whereas in control P*ybx1* embryos *sqt* expression is restricted to a few cells, and *gsc* and *mxtx2* are typically not detected (*Figure 6D*). We found no difference in *lft2, boz, vox* or *vent* expression between M*ybx1* and control embryos (*Figure 6C,D* and data not shown). Thus, early Wnt and FGF signaling targets are not affected in M*ybx1* mutants, whereas expression of many Nodal target genes is precocious and their levels increased.

## The extra-embryonic yolk syncytial layer is expanded in M*ybx1* mutants

YSL expression of mxtx2 is increased in M*ybx1* embryos (*Figure 6C,D*). Accordingly, marginal cells lose cell membranes by early blastula stages, and syncytial nuclei accumulate over the yolk (arrowheads in *Figure 3E* and yellow boxed area in *Figure 6E*). The margin between the blastoderm and YSL, evident by E-cadherin immunolocalization in control embryos, is not clearly demarcated in M*ybx1* embryos. Increased numbers of YSL nuclei (YSN) were observed in M*ybx1*[sa42] and M*ybx1*[sg8] embryos (white arrowheads in *Figure 6E*), and sometimes, nearly 50 YSN were observed at the 512–1000 cell stage. ~50% of M*ybx1* embryos have more than 13 nuclei, whereas control embryos show few or no YSN (*Figure 6F*). The premature formation and increased numbers of YSN leads to substantially fewer cells in the blastoderm, failure to initiate epiboly, and embryonic lethality. These phenotypes are rescued by maternal *ybx1-2a-gfp* transgenes (*Figure 6F*). Thus, the extra-embryonic YSL forms precociously and is expanded in M*ybx1*.

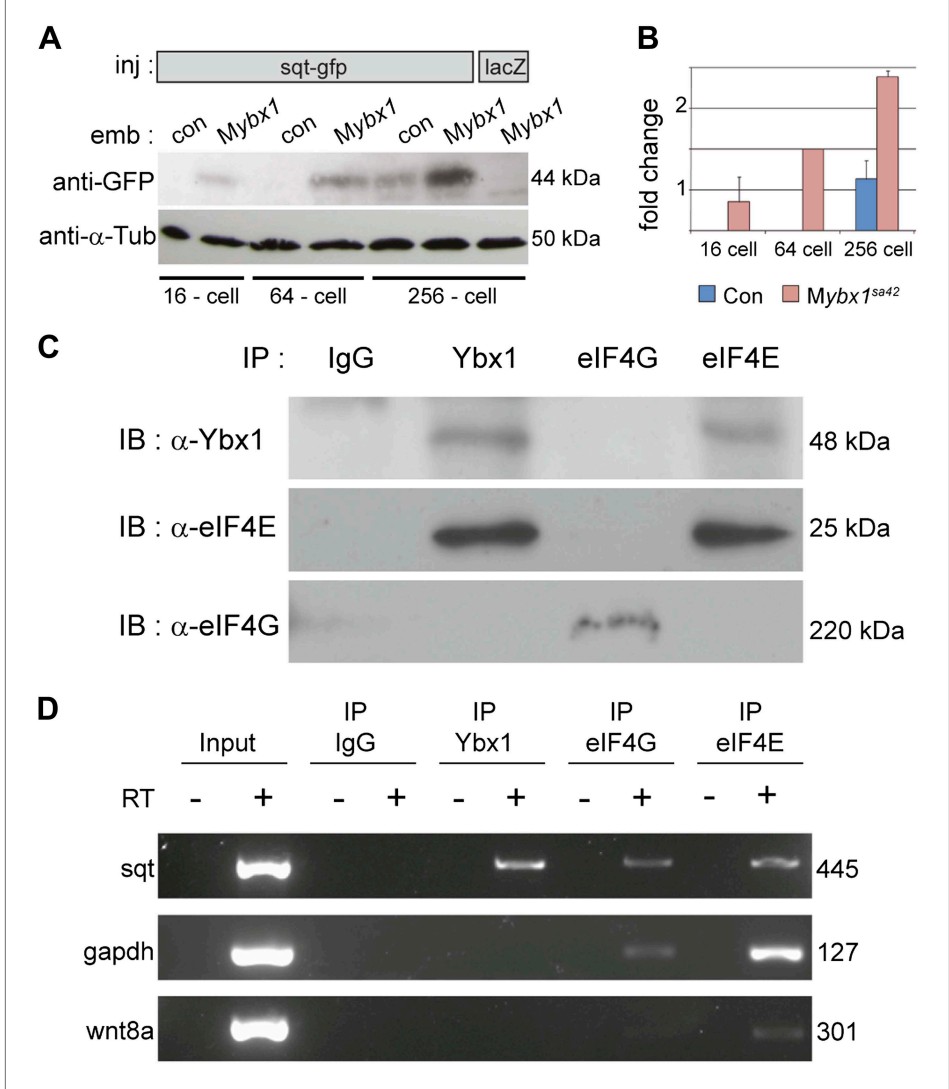

**Figure 5**. Ybx1 interacts with the translation initiation machinery and represses translation of sqt RNA. (**A**) Western blot to detect GFP shows injected sqt-gfp is translated by 16-cells in M*ybx1*, whereas in controls, Sqt-GFP is detected at blastula stages, and lacZ control injection shows no Sqt-GFP. (**B**) Sqt-GFP protein expression is precocious and elevated in M*ybx1* embryos. Error bars in **B** show standard deviation from three experiments. (**C**) Co-immunoprecipitation in embryo lysates followed by western blot analysis shows that Ybx1 interacts with eIF4E. eIF4G binds poorly with Ybx1. Faint smear in control IgG lane for eIF4G is spillover from input lane (see *Figure 5—figure supplement 1C* for complete blot for eIF4G). (**D**) Antibodies towards Ybx1, eIF4G and eIF4E pull down sqt RNA in embryos lysates. RT-PCR on the embryos lysates in panel **C** shows sqt RNA but not control gapdh or wnt8a RNA in RNA-IP with αYbx1 antibodies. Control IgG antibodies do not show any RT-PCR product, whereas antibodies to the translation initiation factor eIF4E can pull down sqt RNA, wnt8a and gapdh RNA, and antibodies to eIF4G detects weak bands for sqt and gapdh in the RNA-IPs. RT-PCR from whole embryo lysates is the positive control. PCR product sizes are indicated on the right.

The following figure supplements are available for figure 5:

**Figure supplement 1**. Translation of control RNAs is not affected in M*ybx1* mutant embryos.

## Nodal signals from the yolk induce and expand the extra-embryonic YSL

Ybx1 is a multi-functional regulator of many target genes. This raises the question of whether the phenotypes observed in M*ybx1* mutants are a direct consequence of Sqt/Nodal translation and diffusion from the yolk in the absence of Ybx1 function, or due to other potential effects of Ybx1. To directly

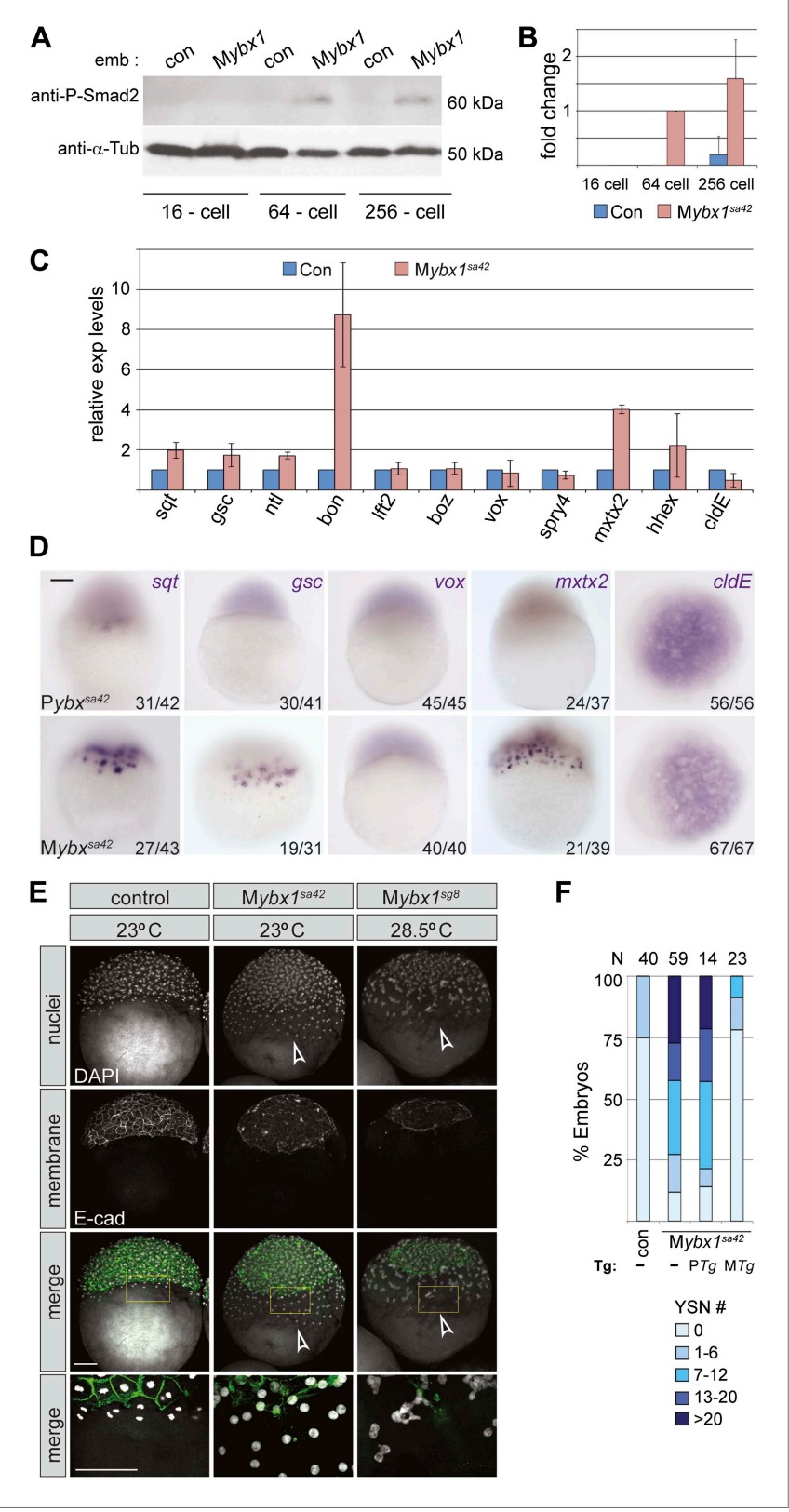

**Figure 6**. Nodal signaling is deregulated in M*ybx1* embryos. (**A**) Phosphorylated-Smad2 is detected at the 64-cell stage in M*ybx1* embryos. (**B**) Phospho-Smad2 levels are elevated in M*ybx1* embryos at cleavage stages, compared to controls. (**C**) Quantitative real-time RT-PCR shows that Nodal target (*sqt, gsc, ntl, bon*) and YSL gene expression
*Figure 6. Continued on next page*

*Figure 6. Continued*

(*mxtx2*) is elevated in M*ybx1* compared to controls, whereas expression of *lefty2*, the Wnt target, *boz*, ventral mesoderm gene *vox*, FGF target *spry4*, and the enveloping layer (EVL) marker *cldE*, is either not significantly altered or marginally reduced. (**D**) Whole mount in situ hybridization shows expanded YSL domains of *sqt, gsc*, and *mxtx2* in M*ybx1* embryos; the *cldE* expression domain is not significantly different from controls, and *vox* is not detectable. Scale bar, 100 µm. (**E**) M*ybx1* mutants have expanded YSL. DAPI staining to detect nuclei and E-cadherin immunostaining to detect membranes shows 1 tier of YSL nuclei (undergoing division) in control embryos. In M*ybx1*^sa42^ and M*ybx1*^sg8^ embryos, multiple tiers of YSL are observed (yellow boxed area). Bottom panels show higher magnification of yellow, boxed area. Cell membranes are clearly demarcated in control embryos, but appear fragmented in M*ybx1* mutants. Scale bars, 100 µm. (**F**) Histogram showing YSL nuclei numbers in M*ybx1* and control embryos, with or without *ybx1* transgenes at 23°C. ~75% of control embryos have no YSL nuclei and only 25% show 1–6 YSL nuclei, whereas ~80% of M*ybx1* embryos and M*ybx1* embryos with P*Tg*, show 7 or more YSL nuclei, and ~25% show >20 YSL nuclei. M*ybx1* embryos with *ybx1* M*Tg* show reduced numbers of YSL nuclei. Number of embryos scored is indicated above the histogram. Error bars in **B** and **C** show standard deviation from three experiments.

determine the effects of excess Nodal protein from the yolk, we implanted Affi-gel beads that were pre-soaked in either control BSA protein or purified mouse Nodal protein, into the yolk of wild-type embryos, and examined YSN (schematic in **Figure 7A**). Bead implantation itself did not disrupt morphogenesis or development (**Figure 7A**). Control BSA bead-implanted embryos showed 1 tier of YSL (with 4–5 nuclei; n = 17, **Figure 7B,C**), similar to wild-type embryos (**Kimmel and Law, 1985**). By contrast, most Nodal bead-implanted embryos showed more YSN (75%, n = 32 embryos; arrowhead in **Figure 7B,C**). Therefore, Nodal diffusion from the yolk is sufficient for YSL expansion. Nodal bead implantation in the yolk of MZ*oep* mutant embryos, which cannot respond to Nodal signals

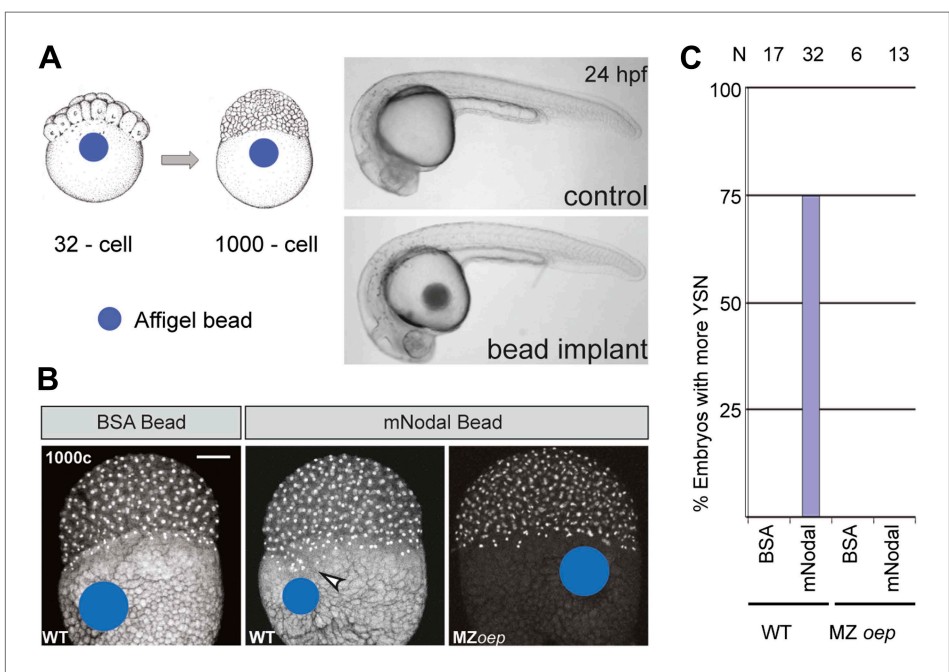

**Figure 7**. Excess Nodal protein from the yolk directly expands the extra-embryonic YSL. (**A**) Affi-gel beads presoaked in either BSA or mNodal were implanted at the 32-cell stage and fixed at 1000-cell stage. BSA-bead implanted embryos develop normally and have no morphological defects at 24 hpf, similar to control embryos. (**B**) DAPI staining shows one tier of YSL nuclei in BSA bead-implanted wild-type embryos whereas mNodal bead-implanted embryos show many nuclei (arrowhead). MZ*oep* embryos with Nodal beads are similar to control BSA bead-implanted embryos. Bead position indicated by blue dot. Scale bar, 100 µm. (**C**) Histogram showing percent wild-type or MZ*oep* embryos with more YSN after bead implants.

(*Schier, 2009*), did not affect the YSL (0%, n = 13; *Figure 7B,C*). Thus, excess Nodal signaling from the yolk directly induces premature and expanded extra-embryonic YSL. Similar YSL expansion and gastrulation defects were reported in lefty-1;lefty-2 double-morphants, where Nodal signaling is deregulated in the absence of the Lefty inhibitors (*Branford and Yost, 2002*; *Feldman et al., 2002*). Taken together, these results suggest that the M*ybx1* phenotypes are likely due to precocious, unregulated and elevated Nodal signaling by de-repression of Sqt translation.

However, expanded YSL could also arise from defects in late cytokinesis during early embryogenesis, leading to aberrant syncytial layer formation (*Yabe et al., 2009*). To distinguish between these possibilities, we blocked Nodal signaling in M*ybx1* mutants by two strategies. Firstly, we injected capped mRNA encoding inhibitor Lefty1 into M*ybx1* embryos to block Nodal signaling. Injections in control P*ybx1* embryos were used to assess the efficacy of the RNA. Immunostaining with antibodies to detect E-cadherin in cell membranes and DAPI staining to detect nuclei show that that the expanded YSL phenotype in M*ybx1* embryos is suppressed by injection of lefty1 RNA, but not with lacZ RNA (*Figure 8A*). The margin between the blastoderm and YSL, which is not clearly demarcated in M*ybx1* embryos, is restored upon injection of lefty1 RNA (*Figure 8A*). The majority of M*ybx1* mutant embryos injected with lacZ RNA fail to initiate gastrulation and none complete gastrulation. In contrast, more than 50% of mutant embryos injected with lefty1 RNA initiate epiboly movements and of these, ~80% complete gastrulation, and survive till prim5 stages (*Figure 8B*). Lefty1 RNA-injected M*ybx1*$^{sa42}$ mutant embryos do not manifest cytokinesis failure or YSL defects even at the restrictive temperature. Therefore, the YSL defects and the failure to initiate gastrulation in M*ybx1* mutant embryos are due to excessive Nodal signaling and not a result of cytokinesis defects.

Secondly, we generated *ybx1*;*sqt* compound mutants. Embryos mutant for the *sqt*$^{cz35}$ allele express maternal sqt RNA that is localized, but the mutant Sqt protein is truncated and non-functional (*Feldman et al., 1998*; *Bennett et al., 2007*; *Lim et al., 2012*). The *sqt*$^{cz35}$ mutation selectively abolishes Sqt signaling without affecting the early functions of maternal sqt RNA or activity of other zebrafish Nodals. We did not recover any *ybx1*;*sqt* double homozygous adults (N > 120 fish), but interestingly, most embryos from *ybx1*$^{sa42}$/*ybx1*$^{sa42}$;*sqt*$^{cz35}$/+ intercrosses, which are essentially M*ybx1* but where some have reduced Sqt or no Sqt, undergo gastrulation at 23°C unlike M*ybx1* single mutants (*Figure 8C,D*). M*ybx1*;Z*sqt* compound mutants show phenotypes typical of reduced Nodal activity such as those observed in MZ*midway* mutant embryos, or upon complete loss of Nodal activity (*Figure 8—figure supplement 1*) (*Thisse et al., 2000*; *Schier, 2009*; *Slagle et al., 2011*). Similar to M*ybx1* embryos injected with lefty RNA, YSL expansion is not observed in M*ybx1*;Z*sqt* mutants (data not shown) and these embryos go through gastrulation. These findings demonstrate that the M*ybx1* mutant phenotypes are a direct consequence of precocious and deregulated maternal Sqt/Nodal signaling.

## Discussion

In this study, we have provided the first direct evidence of translational control of Nodal signaling by a key maternal factor, Ybx1, and demonstrated that it is essential for embryonic development. Our use of a temperature-sensitive *ybx1* allele, that allows selective and conditional disruption of maternal Ybx1 function at early stages, shows that Nodal signaling is the only pathway affected at these stages in the mutants. This allele can potentially be a used as a tool to identify other genes and processes regulated by Ybx1.

Ybx1 is an abundant molecule and neither the RNA nor protein is spatially restricted. How, then, is Ybx1 binding specificity achieved? It is possible that other regions of Ybx1 than the CSD, and other factors in the Ybx1–RNP complex confer specificity to the interactions. Such context-dependent specificity has been observed for many transcription factor complexes and other RNA-binding proteins as well. For example, a bipartite RNA recognition module in Lin28 (which contains a CSD) binds to two distinct regions of let-7 RNA to regulate its biogenesis, and two distinct RNA-binding domains in fragile X mental retardation protein (FMRP), recognize distinct RNA elements (*Tong et al., 2000*; *Nam et al., 2011*). In support of this possibility, we find that in addition to the residues in the CSD, the DD, RNP and ssDBD domains of Ybx1 are also required for binding to sqt RNA, and likely confer specificity to the interactions. By contrast, the CSD-containing Lin28 protein does not bind to the sqt DLE. Furthermore, the *ybx1*$^{sg8}$ mutation, which deletes the Ybx1 C-terminus, which is thought to be a protein interaction domain (*Wolffe, 1994*), results in more severe phenotypes and earlier lethality than *ybx1*$^{sa42}$. It is likely that the residues lacking in the truncated Ybx1$^{sg8}$ peptide are also important for its functions.

Our findings show that a major function of maternal Ybx1 is to regulate Nodal signaling via its effects on sqt RNA localization, processing, and translation. Maternal sqt RNA is largely

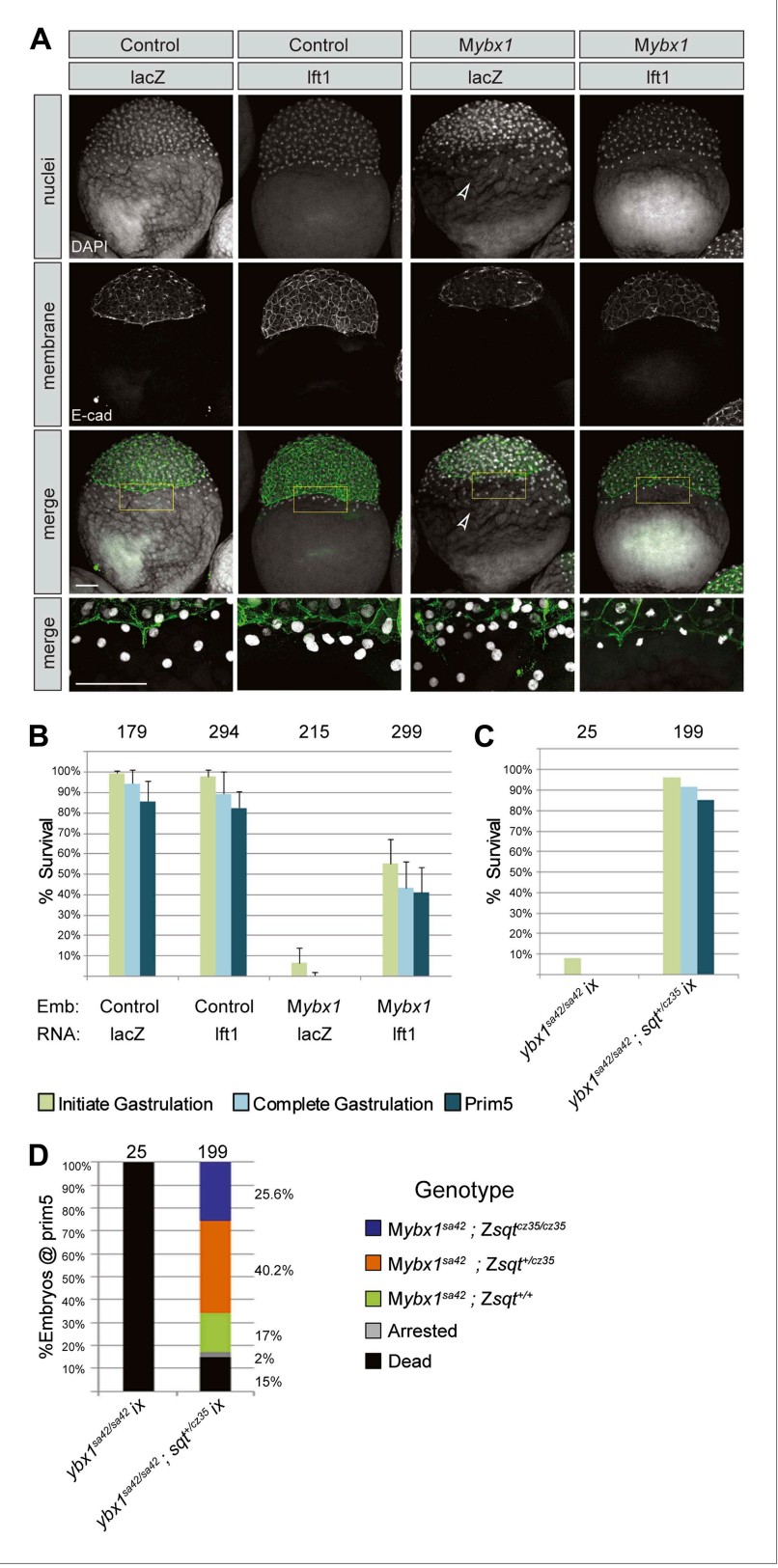

**Figure 8**. YSL and gastrulation defects in M*ybx1* mutants can be rescued by blocking Nodal signaling. (**A**) DAPI-stained nuclei and E-cadherin immunostained membranes are clearly demarcated in lacZ and lefty1 RNA-injected control embryos. E-cadherin staining appears fragmented and number of YSL nuclei is increased in M*ybx1* mutant
*Figure 8. Continued on next page*

*Figure 8. Continued*

embryos injected with control lacZ RNA (white arrowhead). In M*ybx1* embryos injected with lefty1 RNA, the number of YSL nuclei is restored to normal levels and membrane staining does not appear fragmented. Bottom panels show higher magnification of area boxed in yellow; scale bar, 100 μm. (**B**) Histogram showing rescue of gastrulation and survival till prim5 stage by injection of lefty1 RNA but not lacZ RNA in M*ybx1* mutants that were subjected to temperature shift at 23°C. Error bars show standard deviation from three experiments. Number of embryos is shown on top of the histogram. (**C**) Histogram showing gastrulation and % survival in embryos from *ybx1*[sa42/sa42];*sqt*[cz35/+] crosses and M*ybx1* mutants subjected to temperature shift at 23°C. Most embryos from *ybx1*[sa42/sa42];*sqt*[cz35/+] crosses initiate and complete gastrulation in comparison to M*ybx1* [sa42] mutants. (**D**) Histogram showing survival at 23°C and genotypes of embryos from matings of *ybx1*[sa42/sa42];*sqt*[cz35/+], in comparison to M*ybx1*[sa42] embryos which do not survive at 23°C. The expected % for each genotype is 25% for M*ybx1*[sa42];*sqt*[cz35/cz35] and M*ybx1*[sa42];*sqt*[+/+], and 50% for M*ybx1*[sa42];*sqt*[cz35/+]. All M*ybx1*[sa42];*sqt*[cz35/cz35] embryos (which have no Sqt signaling) survive, whereas many M*ybx1*[sa42];*sqt*[cz35/+] and M*ybx1*[sa42];*sqt*[+/+] do not survive at 23°C. Number of embryos scored is indicated above the histogram, and % observed for each genotype is indicated to the right of the colored bars.

The following figure supplements are available for figure 8:

**Figure supplement 1**. M*ybx1*;Z*sqt*[cz35/cz35] mutant embryos show phenotypes typical of reduced Nodal activity.

non-polyadenylated at early cleavage stages, and polyA-containing sqt RNA is normally detected from the 16-cell stage (*Lim et al., 2012*). In M*ybx1* embryos, polyA-sqt RNA is detected even at the 1-cell stage, indicating premature polyadenylation. Regulating the length of the polyA tail is known to mediate translational control of the RNA; for example, cyclin A RNA is stored in *Drosophila* oocytes with short polyA tails (*Morris et al., 2005*). Ybx1 interacts with the translation initiation factor eIF4E and the sqt 3′UTR. Interactions of 3′UTR binding proteins with translation initiation factors, such as the CPEB-Maskin-eIF4E complex, has been shown in translational control of maternal RNAs in *Xenopus* oocytes. Maskin binds the cap-binding factor eIF4E, and prevents interactions between eIF4G and eIF4E, which is required for recruitment of the 40S ribosome subunit to the 5′end of mRNAs, and thereby represses translation (*Cao and Richter, 2002*). Similarly, in *Drosophila* oocytes, cup binds eIF4E and Bruno to regulate oskar RNA translation (*Nakamura et al., 2004*). Mammalian YB1 prevents eIF4G from binding to eIF4E, and blocks initiation of translation (*Nekrasov et al., 2003*). Binding of Ybx1 to the sqt 3′UTR and eIF4E likely prevents eIF4G and eIF4E complex formation, and in M*ybx1* mutants, in the absence of Ybx1 function, Sqt translation occurs precociously. Thus, Ybx1 binding to the translation initiation factors and the sqt 3′UTR can lead to translational repression of sqt RNA.

Consistent with Ybx1 being a DLE-binder, sqt RNA localization is disrupted in M*ybx1*, and many Sqt/Nodal target genes (including sqt) show precocious and elevated expression. Surprisingly, the Nodal target *lefty2* is not detected in mutant embryos. Therefore, *lefty2* expression requires factors or inputs that are missing in M*ybx1*. The lack of feedback inhibition in the absence of Lefty2 together with elevated Sqt/Nodal levels likely exacerbates deregulated Nodal signaling in M*ybx1*. The YSL expansion and gastrulation defects in M*ybx1* mutants can be rescued by blocking Nodal signaling via lefty overexpression or by using the *sqt*[cz35] genetic background that lacks the signaling functions of Sqt, indicating that these phenotypes arise from excess Nodal signaling. Interestingly, M*ybx1*;Z*sqt* compound mutant embryos are similar to *cyc;sqt* double mutants (*Feldman et al., 1998*), suggesting that maternal Ybx1 may have additional functions in regulation of Nodal signaling. We also found that Wnt signaling targets are not induced in M*ybx1* mutants, where wnt8a RNA localization and Wnt8a-GFP protein expression are normal, but maternal sqt RNA is mis-localized. Therefore, the response to the maternal Wnt signal requires dorsal localization of maternal sqt RNA. This supports our previous findings, where over-expression of localized non-coding sqt RNA increased dorsal β-Catenin nuclei numbers and elevated Wnt target gene expression, and mis-localization of sqt RNA by morpholinos that also block Sqt translation resulted in loss of dorsal β-Catenin accumulation (*Lim et al., 2012*).

We had previously shown that human NODAL 3′UTR fused with lacZ localizes dorsally in zebrafish (*Gore et al., 2005*). This was surprising since mammalian embryos are thought to undergo regulative development. Moreover, Nodal RNA is not localized in early mouse embryos (*Robertson et al., 2003*; Cheong and Sampath, unpublished observations). Ybx1 binds the DLE, and regulates localization and translation of sqt. The sqt DLE, therefore, encompasses a localization and translational control element.

This suggests that the mammalian NODAL 3'UTR may also harbor a translational control module. Our finding that maternal sqt/nodal must be translationally repressed, and that deregulated maternal Nodal signaling is catastrophic, shows that this is an essential control mechanism. Translational control is a new paradigm in regulation of this pathway. It will be interesting to see if this mechanism regulates Nodal signaling in other organisms or biological processes. Since human NODAL and ALK7 receptors are expressed in the ovary and placenta, and elevated NODAL is associated with pre-eclamptic placentas (*Nadeem et al., 2011*), precise regulation of maternal Nodal signaling is likely to be important for human placentation. Moreover, uncontrolled and deregulated Nodal signaling has been associated with metastasizing tumors, underscoring the importance of precise and timely regulation of Nodal signaling. Finally, Nodal signaling is essential for maintaining stem cell pluripotency, and current methods to generate and maintain embryonic stem cell (ESC) and induced pluripotent stem cells (iPSC) rely upon transcription factors. Our finding that Nodal signaling is maternally regulated by translational repression could allow modulation of these important therapeutic cells by this new mechanism.

## Materials and methods

### RNA gel-shifts and UV cross-linking assays

Embryos were homogenized in 1/10 vol lysis buffer (20 mM Tris pH 8.0, 100 mM NaCl, 0.1 mM EDTA, 1 mM 6-aminohexanoic acid, 1 mM PMSF, 25% glycerol) to make extracts. Debris was pelleted by centrifugation ($20,000 \times g$, 4°C, 1 min), and supernatants flash frozen in 50 µl aliquots in liquid $N_2$. 100 nucleotide long probes spanning the 3'UTR of sqt, wnt8a (*Lu et al., 2011*) and vg1 (*Bally-Cuif et al., 1998*) were synthesized and used in RNA gel-shift assays. Templates for the probes were generated by PCR with an extended phage T3 RNA polymerase promoter (AATTAACCCTCACTAAAGGGAGAA) appended to the 5'end of the 5'primer, and gel-purified. Primers are listed in *Supplementary file 1*. Radioactively labeled probes were transcribed with T3 RNA polymerase (Promega, Madison, WI), mixed with extracts, and used in electrophoretic mobility-shift assays. For the competition gel-shift assays ~0.1 ng of radioactive probe was competed with 5–80 ng of unlabeled RNA. RNA cross-linking reactions were essentially the same as RNA gel-shifts, except that the reactions were UV-cross-linked for 5 min in a Stratalinker (Stratagene, La Jolla, CA), digested with RNase A (0.5 µg) for 1 hr at 37°C, and separated on an SDS-PAGE gradient gel (7%, 29:1 acrylamide:bisacrylamide to 12%, 19:1 acrylamide:bisacrylamide) at ~5 mA/1 mm gel overnight, dried, and auto-radiographed.

### Protein purifications

Extracts were made as above, and flash frozen in 2 ml aliquots. Chromatography was performed on an Akta purifier (GE Healthcare, Little Chalfont, UK). 200–500 mg of protein extract was injected through a 0.2 µm syringe filter (Minisart; Sartorious, Göttingen, Germany) to a pre-equilibrated heparin HiTrap column (GE Healthcare) and eluted with a $(NH_4)_2SO_4$ gradient. Fractions were collected and assayed by gel-mobility shift with sqt1 probes. Positive fractions were pooled and loaded onto coupled octyl sepharose and phenyl sepharose columns. In the conditions used, SRBF1 passes through octyl sepharose and binds to the phenyl sepharose column. The columns were uncoupled, and SRBF1 was eluted from the phenyl sepharose column with a $(NH_4)_2SO_4$ gradient. Positive fractions were pooled, dialyzed, and loaded onto a 1 ml heparin HiTrap column (GE Healthcare), eluted with a NaCl gradient, collecting 1 ml fractions. We used 1–5 µl of each fraction for gel-shifts or RNA cross-linking assays. Fractions were concentrated and loaded on an SDS-PAGE gradient gel. The gel was stained with colloidal Coomassie blue (*Kang et al., 2002*) and the 48 kDa band was excised and analyzed by mass spectrometry.

### Generation of constructs

The coding sequence of *ybx1* was amplified by PCR (with primers including restriction sites, for *Nco*I and *Bam*HI or *Bgl*II) from zebrafish ovary or embryo cDNA, restriction digested, and cloned into pTrcHISa. Mutations were made by site-directed mutagenesis (*Zheng et al., 2004*). Template plasmid was amplified by PCR with partially overlapping forward and reverse primers (*Supplementary file 1*) using Vent Polymerase (NEB, Ipswich, MA), digested with *Dpn*I, and transformed into XL1blue cells.

### Generation of *ybx1* mutant fish

Libraries of ENU-mutagenized zebrafish were screened for point mutations in the coding region of *ybx1* (*Winkler et al., 2011*). A region encompassing exons two to four of zebrafish *ybx1* (chromosome 8: 49299968 to 49308225; Ensembl Zv9) was amplified by nested PCR using primers listed in

*Supplementary file 1*. Sanger sequencing of PCR fragments was performed with universal M13 forward sequencing primer. Primary hits were amplified and re-sequenced independently and verified. Mutant *ybx1sa42* zebrafish (which harbor a V83F amino acid substitution) were propagated further and bred to homozygosity. For generating deletions in *ybx1* we used a pair of zinc finger nucleases recognizing exon 5 of *ybx1* (Toolgen Inc., Seoul, South Korea) (*Doyon et al., 2008*; *Meng et al., 2008*). Capped mRNA was synthesized from linearized plasmids, and 25 pg RNA of each zinc finger nuclease pair was injected in 1-cell-stage AB wild-type embryos. Injected embryos were raised to adulthood and progeny screened for mutations in the *ybx1* locus by PCR and sequencing. We identified several small deletions at the target site. The *ybx1sg8* allele used in this study has a 5-nucleotide deletion in exon 5 of *ybx1*, which leads to a frame-shift after amino acid 197 and premature termination after amino acid 205.

## Zebrafish strains

Wild-type, *ybx1sa42*, *ybx1sg8*, *sqtcz35* and *oeptz57* fish were maintained at 28.5°C, and embryos obtained by natural mating using standard procedures, in accordance with institutional animal care regulations (*Westerfield, 2007*). Embryos from homozygous *ybx1sa42* females were collected, incubated at 28.5°C until the first cell division, and then shifted to 23°C for observing the temperature-sensitive pheno-type. A few homozygous *ybx1sa42* females yield embryos that manifest a range of phenotypes, some of which survive at 23°C. In this study, homozygous *ybx1sa42* females that yielded 100% embryos arrested at gastrula stages were used in all experiments. Embryos from homozygous *ybx1* males and wild-type females (P*ybx1*), are indistinguishable from wild-type embryos, and were used as controls. For examin-ing *ybx1;sqt* double mutant phenotypes, embryos from matings of *ybx1sa42/sa42;sqtcz35/+* fish were incu-bated at 28.5°C until the 4-cell stage to allow sqt RNA localization, shifted to 23°C until the 128-cell stage, and subsequently returned to 28.5°C until observation at gastrula and prim5 stages. The geno-types of mutants were determined by PCR using primers listed in *Supplementary file 1*.

## Lefty RNA injections

Capped synthetic lefty1 RNA was synthesized from linearized plasmid using the mMessage mMachine SP6 kit (Invitrogen, Carlsbad, CA). 2 pg aliquots of lefty1 RNA were injected into M*ybx1sa42* mutant or P*ybx1sa42* control embryos at the 1-cell stage. Capped lacZ RNA was injected as a control. The embryos were incubated at 28.5°C until the 4-cell stage to allow sqt RNA localization, shifted to 23°C until the 256-cell stage, and subsequently returned to 28.5°C until observations at gastrula and prim5 stages.

## Generation of *ybx1* rescue transgene

An 8.26 kb *ybx1* genomic fragment was amplified by PCR, fused with the viral peptide 2a and gfp sequences, cloned into pMDs6 plasmid and co-injected with Ac II transposase mRNA into *ybx1sa42* embryos at the 1-cell stage (*Emelyanov et al., 2006*). Injected embryos were raised to adulthood, and progeny were screened for GFP expression. Two independent *Tg(ybx1-2a-gfp)* transgenic lines were used in this study.

## RNA immunoprecipitation

RNA-IP was carried out using embryos lysates (*Niranjanakumari et al., 2002*). 20 mpf embryos were cross-linked in formaldehyde, and lysed. Anti-Ybx1 (4F12, Sigma, St. Louis, MO), anti-eIF4G (#2469, Cell Signaling Technology, Danvers, MA) and anti-eIF4E (#2067, Cell Signaling Technology) antibodies were bound to protein A/G beads (Calbiochem, EMD Millipore, Billerica, MA), incubated with wild-type embryo lysate at 4°C, washed, and eluted. Half of the eluate was used to detect proteins by western blot and the remainder was used for RNA extraction using TRIzol reagent (Invitrogen), followed by RT-PCR to detect sqt, wnt8a, and gapdh (primer details in *Supplementary file 1*).

## Protein expression and detection

*E. coli* BL21 cells were transformed with plasmids encoding wild-type and mutant Ybx1. Expression of recombinant protein in lysates was detected by Western blots with an anti-6xHis antibody (1:2500; sc50973, Santa Cruz Biotechnology Inc., Dallas, TX), and equal amounts of *E. coli* lysates were used in gel-shift assays. To detect Sqt translation, P*ybx1sa42* and M*ybx1sa42* embryos were injected with 20 pg sqt-GFP RNA. Whole embryo lysates (50 µg) were separated on an 8% SDS-PAGE gel, transferred to High bond-C Extra Membrane (GE Healthcare), and immunoblotting was performed using anti-GFP primary antibodies (1:2500; ab290, Abcam, Cambridge, UK) and HRP-conjugated anti-rabbit IgG secondary antibodies (1:10,000; DAKO, Glostrup, Denmark). Endogenous phospho-Smad2 was

detected using anti-pSmad2 primary antibodies (1:1000; #3101, Cell Signaling Technology), and HRP-conjugated anti-rabbit IgG secondary antibodies (1:5000; DAKO). Endogenous Ybx1 expression in embryos was detected using a mouse anti-Ybx1 antibody (1:1000; 4F12, Sigma), and HRP-conjugated anti-mouse IgG secondary antibody (1:10,000; DAKO). Anti-eIF4E (1:2000; #2067, Cell Signaling Technology) and anti-eIF4G (1:2000; #2469, Cell Signaling Technology) antibodies were used in co-immunoprecipitation assays and western blots to detect interactions with Ybx1.

### Semi-quantitative and quantitative RT-PCR

Total RNA was extracted from embryos using TRIzol reagent (Invitrogen), and 500 ng RNA from WT, P*ybx1*$^{sa42}$ or M*ybx1*$^{sa42}$ embryos was used for first-strand cDNA synthesis. PCR reactions were performed as described (*Lim et al., 2012*) using primers listed in *Supplementary file 1*.

### Whole mount in situ hybridization

1-cell and 4-cell stage embryos were fixed in buffer containing 4% paraformaldehyde, 4% sucrose, and 120 µM calcium chloride in 0.1M phosphate buffer (pH 7.2). Blastula stage embryos were fixed in 4% paraformaldehyde/PBS. Fixed embryos were processed for whole mount in situ hybridization (*Tian, 2003*) to detect *claudinE, cyclinb1, eomesodermin, goosecoid, mxtx2, squint, vasa, vox, wnt8a*, and *ybx1* expression (*Stachel et al., 1993*; *Yoon et al., 1997*; *Howley and Ho, 2000*; *Melby et al., 2000*; *Gore et al., 2005*; *Siddiqui et al., 2010*; *Hong et al., 2011*; *Lu et al., 2011*; *Du et al., 2012*; *Lim et al., 2012*).

### Membrane and nuclear staining

We used anti-E-cadherin antibodies to detect cell membrane adhesions. Control or mutant embryos at the 1000-cell stage were fixed in 4% paraformaldehyde/PBS, and processed for fluorescence immunohistochemistry using rabbit polyclonal anti-E-cadherin antibodies (gift from CP Heisenberg) and Alexa-488-conjugated goat anti-rabbit secondary antibodies (Molecular Probes, Eugene, OR). For detecting nuclei, embryos were fixed with 4% parafomaldehyde/PBS, washed in PBS containing Tween-20 (PBST), incubated with 500 pg/ml DAPI, and washed with PBST. To label yolk syncytial nuclei in live embryos, 4 nl of 0.5 mM SYTOX orange (Invitrogen) was injected into the yolk of 64-cell stage embryos. Labeled nuclei were scored at 512-1K cell stages.

### Bead implantation

Affi-Gel blue beads (50–100 mesh; Bio-Rad Laboratories Inc., Hercules, CA) were pre-soaked in Bovine Serum Albumin (BSA; 100 µg/ml; NEB, Ipswich, MA) or mouse Nodal protein (125-250 µg/ml; R&D systems, Minneapolis, MN) for 30 min. Single Affigel beads were implanted into the yolk of dechorionated 32-cell stage embryos by making a small incision in the yolk with a tungsten needle, and nudging the bead into the yolk with pair of fine forceps. For DAPI or SYTOX staining, implanted embryos were cultured in 30% Danieau's buffer, and processed as described above.

## Acknowledgements

We thank the Sampath laboratory, Mohan Balasubramanian, Tom Carney, Steve Cohen, Ray Dunn, Iain Drummond, Aniket Gore, Greg Jedd, Katsutomo Okamura, and Pernille Rorth for comments and suggestions; *EU FP6 Integrated Project: Zebrafish Models for Human Development and Disease*, and the Sanger Institute Zebrafish Mutation Resource for *ybx1*$^{sa42}$; Bruno Reversade for mNodal protein; Carl-Philipp Heisenberg and Ong Sin Tiong for antibodies; TLL facilities; Helen Quach, Cherish Tay, and Quek Mei Xing for technical assistance. The TILLING facility of the MPI CBG/CRT Dresden and the Sanger Institute Zebrafish Mutation Resource were supported by the ZF-MODELS Integrated Project (contract number LSHG- CT-2003-503496 of the European Commission). The Sanger Institute Zebrafish Mutation resource is also sponsored by the Wellcome Trust (grant number WT 077047/Z/05/Z).

## Additional information

### Funding

| Funder | Author |
| --- | --- |
| Temasek Life Sciences Laboratory | Pooja Kumari, Patrick C Gilligan, Shimin Lim, Karuna Sampath |
| Singapore Millenium Foundation | Patrick C Gilligan, Shimin Lim |

| Funder | Author |
|---|---|
| Nanyang Technological University | Shimin Lim |
| Mechanobiology Institute, National University of Singapore | Long Duc Tran |
| Agency for Science, Technology, and Research | Robin Philp |
| Max Planck Society | Sylke Winkler |

The funders had no role in study design, data collection and interpretation, or the decision to submit the work for publication.

## Author contributions

PK, PCG, KS, Conception and design, Acquisition of data, Analysis and interpretation of data, Drafting or revising the article; SL, Acquisition of data, Analysis and interpretation of data, Drafting or revising the article; LDT, RP, Acquisition of data, Analysis and interpretation of data; SW, Acquisition of data, Analysis and interpretation of data, Contributed unpublished essential data or reagents

## Ethics

Animal experimentation: This study was performed in strict accordance with the recommendations of the Institutional Animal Care and Use Committee at the Temasek Life Sciences Laboratory, Singapore.

# Additional files

## Supplementary files

• Supplementary file 1. Primer sequences.

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
