## [Decision Letter]

Thank you for choosing to send your work entitled “An essential role for maternal control of nodal signaling” for consideration at *eLife*. Your article has now been peer reviewed and we regret to inform you that your work will not be considered further for publication at this time. Your submission has been evaluated by 3 reviewers, one of whom is a member of our Board of Reviewing Editors, and a Senior editor, and the decision was reached after discussions between the reviewers. Although the work is of interest, we regret to inform you that the consensus of the reviewers is that the findings at this stage are too preliminary for publication by *eLife*.

The following individual responsible for the peer review of your submission has agreed to reveal her identity: Ruth Lehmann, Reviewing editor.

While this is an extensive analysis starting with an RNA binding sequence, identifying a transacting factor and finally connecting this transacting factor to the regulation of the target RNA, the results have to be considered in view of what we learn new about Ybx1 protein and the developmental process that is addressed. For example, it is important to note that Ybx1 has had several different names over the years including p54 and Frgy and since it's discovery in the early ’80s (if not before) has been characterized as a fairly ubiquitous sequence-nonspecific RNA binding protein of relatively unclear function. Some authors liken it to be an “RNA histone” whereas others think that it stabilizes mRNA by binding RNA in a cap-dependent manner. It is also a transcription factor and binds a “Y box.” Thus the protein has a long and checkered history and, like TAR RNA binding protein, often is identified in screens because of its abundance and its apparent ability to bind nucleic acids with little specificity. Thus all reviewers feel that additional experimental evidence would need to be provided to make the paper suitable for *eLife*.

The major points that would need to be addressed are:

1) More direct evidence for a specific molecular role of Ybx1 in squint regulation. Otherwise this study will not go beyond the many other papers on Ybx1 and will just be another description of Ybx1 binding to RNA. Here are some suggestions: A) provide quantitative binding assays that demonstrate specificity. B) Provide evidence regarding the mechanism for how Ybx1 inhibits translation. Indeed, there is no convincing direct evidence in this manuscript that Ybx1 really does regulate squint mRNA translation, this is important given this protein's long checkered history. Also see for example evidence for Ybx1 promoting rather than repressing translation in cancer cells during EMT (Evdokimova, et al., (2009). Or C) provide evidence supporting the mechanisms by which Ybx1 suppresses splicing of sqt RNA in the cytoplasm. This is an intriguing observation, but evidence for “cytoplasmic splicing” is at best debatable (see König et al., (2007) versus Steitz et al. (2008)). Data provided in the manuscript are also compatible with a role of Ybx1 in transcription.

You would need to specifically address any of these points and provide novel insight into the function of Ybx1 for any further consideration.

2) You conclude that *sqt* is the major target of Ybx1. The maternal-effect phenotype of Ybx1 is interpreted as an expansion of the YSL. A functional correlate via Nodal/Sqt signaling is provided by the finding that Nodal beads implanted into wild-type embryos affect YSL but not when transplanted into oep mutants. However, the phenotype can also be interpreted as a defect in late cytokinesis during early embryogenesis, leading to an aberrant ectopic syncytium. Mybx1^sg8^ mutants exhibit rounded cells as well as loose cells in the chorionic space (Figure 3), which are very characteristic of defects in late cytokinesis (sometimes embryos with these defects also become overtly abnormal at about the third cell cycle). The later defect in Mybx1^sa42^ is also characteristic of weaker versions of these phenotypes. Although the manuscript concludes that the early cleavage stage defects in M*ybx* mutants is caused by an enlarged YSL, the possibility that these defects are simply a cause of an aberrant syncytium due to defects in early cytokinesis is not well substantiated. Similarly, while it is plausible that excess nodal signaling is behind the overt “syncytial” phenotypes of the M*ybx1* mutants, an alternative possibility is that targets other than nodal are also misregulated, resulting in defects in cell division. The transplanted bead experiments do not preclude this possibility. A crucial and absolutely critical experiment to test this would be a M*ybx1*, MZ*oep* double mutant.

3) The most apparent early defect observed in M*ybx1* mutants is that of mRNA transport along what appear to be axial streamers, from the ooplasm in the yolk to the blastodisc where the mRNA becomes localized. Is this the primary phenotype, rather than the proposed precocious translation? Is the effect truly specific to transport of maternal nodal mRNA? Figure 4—figure supplement 1 A states that Wnt8a transport can also be affected. It may be that the defect mostly affects ooplasmic transcripts that need to be transported to the forming blastodisc. Do the control mRNAs encompass all known pathways of mRNA localization, in particular those transported animally through axial streaming? If the M*ybx1*, MZ*oep* double mutant shows a clear *oep* phenotype, these additional hypotheses should be at least discussed in the text.

---

## [Author Response]

We thank the editor and reviewers for their comments. We have performed new experiments, and show the new data in revised Figures 1, 2, 3, 5, 6, and new Figure 8, and revised figure supplement files for Figures 1, 2, 4, 5, and new Figure 8—figure supplement 1.

*While this is an extensive analysis starting with an RNA binding sequence, identifying a transacting factor and finally connecting this transacting factor to the regulation of the target RNA the results have to be considered in view of what we learn new about Ybx1 protein and the developmental process that is addressed. For example, it is important to note that Ybx1 has had several different names over the years including p54 and Frgy and since it's discovery in the early ’80s (if not before) has been characterized a fairly ubiquitous sequence-nonspecific RNA binding protein of relatively unclear function. Some authors liken it to be an “RNA histone” whereas others think that it stabilizes mRNA by binding RNA in a cap-dependent manner. It is also a transcription factor and binds a “Y box.” Thus the protein has a long and checkered history and, and like TAR RNA binding protein, often is identified in screens because of its abundance and its apparent ability to bind nucleic acids with little specifically. Thus all reviewers feel that additional experimental evidence would need to be provided to make the paper suitable for* eLife.

*The major points that need to be addressed are*:

*1) More direct evidence for a specific molecular role of Ybx1 in squint regulation. Otherwise this study will not go beyond the many other papers on Ybx1 and will just be another description of Ybx1 binding to RNA. Here are some suggestions: A) provide quantitative binding assays that demonstrate specificity. B) Provide evidence regarding the mechanism for how Ybx1 inhibits translation. Indeed, there is no convincing direct evidence in this manuscript that Ybx1 really does regulate squint mRNA translation, this is important given this protein's long checkered history. Also see for example evidence for Ybx1 promoting rather than repressing translation in cancer cells during EMT (Evdokimova, et al., (2009). Or C) provide evidence supporting the mechanisms by which Ybx1 suppresses splicing of sqt RNA in the cytoplasm. This is an intriguing observation, but evidence for “cytoplasmic splicing” is at best debatable (see König et al, (2007) versus Steitz et al. (2008)). Data provided in the manuscript are also compatible with a role of Ybx1 in transcription*.

*You would need to specifically address any of these points and provide novel insight into the function of Ybx1 for any further consideration*.

We agree that direct evidence of Ybx1 regulation of sqt is essential to gain fresh insight into Ybx1 functions. Towards this end, we have performed new experiments to examine how Ybx1 regulates sqt translation, to address point 1B.

Specifically, we determined if Ybx1 forms a complex with the translation initiation machinery and sqt RNA.

As shown in revised Figure 5 and revised Figure 5—figure supplement 1, *Ybx1 interacts with eIF4E in the translation initiation complex*. eIF4E is known to bind the 5 methyl guanosine cap at the 5’UTR of mRNAs. Furthermore, RNA-IP with Ybx1, eIF4G and eIF4E antibodies pulls down sqt RNA (see revised Figure 5) whereas control RNAs (gapdh and wnt8a) interact with the eIF4G and eIF4E proteins, but not with Ybx1. Thus, Ybx1 interacts with the translation initiation machinery and with the DLE in the sqt 3’UTR.

Response to point 1A: to demonstrate specificity of Ybx1 interactions with sqt RNA, we show that:

1) Deletions in the DLE that disrupt sqt dorsal localization also abolish Ybx1 binding.

2) Ybx1 does not bind the DLE antisense probe.

3) Other localized RNAs do not compete for binding in semi-quantitative competition gel-shift assays with 5 to 80-fold excess of cold competitor RNAs (see revised Figure 1, where we show that sqt RNA with the sqt 3’UTR binds preferentially to SRBF1/Ybx1).

4) With antibodies to Ybx1 we get RNA-IP on sqt, and not other RNAs.

5) In contrast to Ybx1, cold shock domain containing proteins such as Lin28A do not bind sqt (see revised Figure 2—figure supplement 2).

Therefore, although it is surprising, taken together, the evidence we have provided in our manuscript supports specific binding of Ybx1 to the sqt DLE.

Response to point 1C: The interesting link between splicing of sqt RNA and ybx1 will require additional work, particularly in light of the cytoplasmic splicing literature, and therefore we feel this issue is beyond the scope of the present Kumari et al. manuscript.

*2) You conclude that* sqt *is the major target of Ybx1. The maternal-effect phenotype of Ybx1 is interpreted as an expansion of the YSL. A functional correlate via Nodal/Sqt signaling is provided by the finding that Nodal beads implanted into wild-type embryos affect YSL but not when transplanted into oep mutants. However, the phenotype can also be interpreted as a defect in late cytokinesis during early embryogenesis, leading to an aberrant ectopic syncytium. Mybx1sg8 mutants exhibit rounded cells as well as loose cells in the chorionic space (*Figure 3*), which are very characteristic of defects in late cytokinesis (sometimes embryos with these defects also become overtly abnormal at about the third cell cycle). The later defect in Mybx1sa42 is also characteristic of weaker versions of these phenotypes. Although the manuscript concludes that the early cleavage stage defects in* M*ybx mutants is caused by an enlarged YSL, the possibility that these defects are simply a cause of an aberrant syncytium due to defects in early cytokinesis is not well substantiated. Similarly, while it is plausible that excess nodal signaling is behind the overt “syncytial” phenotypes of the* M*ybx1 mutants, an alternative possibility is that targets other than nodal are also misregulated, resulting in defects in cell division. The transplanted bead experiments do not preclude this possibility. A crucial and absolutely critical experiment to test this would be a* M*ybx1,* MZ*oep double mutant. “If the* M*ybx1,* MZ*oep double mutant shows a clear oep phenotype, these additional hypotheses should be at least discussed in the text”*.

Indeed, the most direct evidence that the phenotypes observed in M*ybx1* mutants are due to excess Sqt/Nodal signaling is to examine *ybx1/nodal* mutants.

To test this, we generated *ybx1;oep* and *ybx1;sqt*^*cz35*^ mutant lines and attempted to generate double maternal mutants. Embryos mutant for the *sqt*^*cz35*^ allele express maternal sqt RNA that is localized (37), but the truncated Sqt protein is non-functional (19; 2; 37). The *sqt*^*cz35*^ mutation selectively abolishes Sqt/Nodal signaling without affecting the early localization and functions of maternal sqt RNA (37) and thus, is a better tool to address this question than *oep* mutants that also lack signaling from the other zebrafish nodals, *cyc* and *spaw*.

We did not recover any M*ybx1*;M*sqt* homozygous adults (from >120 progeny screened), but interestingly, embryos from matings of *ybx1*^*sa42*^/*ybx1*^*sa42*^;*sqt*^*cz35*^/+ parents (which are essentially M*ybx1* but in which some have reduced *sqt* or no *sqt*) yield embryos with phenotypes identical to nodal mutants or lefty/antivin over-expression. Importantly, unlike M*ybx1* single mutants, the M*ybx1;sqt*^*cz35*^ compound mutant embryos can initiate and complete gastrulation, and do not manifest any cytokinesis or YSL defect even at the restrictive temperature (new Figure 8). Interestingly, in these crosses, the embryos that do not survive are those with at least one wild-type copy of *sqt* (new Figure 8).

We also blocked Nodal signaling in M*ybx1* mutants by injecting lefty1 inhibitor RNA and find that the injected mutants undergo normal cytokinesis and gastrulation, unlike mutant embryos injected with control lacZ RNA. We show this data in new Figure 8.

Thus, the genetic evidence from M*ybx1;Zsqt* mutants and over-expression experiments with lefty1 indicate that the early gastrula and YSL defects in M*ybx1* can be directly ascribed to excess Sqt/Nodal signaling, and are not due to cytokinesis or other defects.

*3) The most apparent early defect observed in* M*ybx1 mutants is that of mRNA transport along what appear to be axial streamers, from the ooplasm in the yolk to the blastodisc where the mRNA becomes localized. Is this the primary phenotype, rather than the proposed precocious translation? Is the effect truly specific to transport of maternal nodal mRNA?*
Figure 1—figure supplement 1
*states that Wnt8a transport can also be affected. It may be that the defect mostly affects ooplasmic transcripts that need to be transported to the forming blastodisc. Do the control mRNAs encompass all known pathways of mRNA localization in particular those transported animally through axial streaming? If the* M*ybx1,* MZ*oep double mutant shows a clear oep phenotype, these additional hypotheses should be at least discussed in the text*.

We examined several localized RNAs representing various classes of transport (cortical, axial streamer, vegetal etc.). As shown in Figure 4 and revised Figure 4—figure supplement 1, expression of vasa, cyclin B1, eomesodermin and wnt8a as representative examples is unaffected in M*ybx1* mutant embryos. We did not observe any defects in vegetal localization of grip2 RNA in M*ybx1*^*sa42*^. Expression of cyclin B1 and snail1a, which are thought to be transported by axial streamers, is also unaffected in the mutants.

Even for wnt8a, while we observe a slight lag in a minority of M*ybx1*^*sg8*^ deletion mutant embryos, wnt8a RNA does in fact get transported and is correctly localized both to the blastoderm and asymmetrically to the vegetal cortex (black arrowheads and arrow in revised Figure 4—figure supplement 1), by the 4-cell stage in all mutant embryos. This slight lag is not observed in M*ybx1*^*sa42*^ even at the restrictive temperature.

Therefore, amongst the maternal RNAs (representative of the various transport pathways) examined to date, we only observe a defect in maternal sqt RNA localization.